# Real-time observation of a metal complex-driven reaction intermediate using a porous protein crystal and serial femtosecond crystallography

Basudev Maity [1] ✉, Mitsuo Shoji [2] ✉, Fangjia Luo[3], Takanori Nakane [4], Satoshi Abe[1], Shigeki Owada[3,5], Jungmin Kang [5], Kensuke Tono [3,5], Rie Tanaka [5,6], Thuc Toan Pham [1], Mariko Kojima [1], Yuki Hishikawa [1], Junko Tanaka[1], Jiaxin Tian[1], Misaki Nagama[1], Taiga Suzuki[1], Hiroki Noya[1], Yuto Nakasuji[1], Asuka Asanuma[1], Xinchen Yao[1], So Iwata[5,6], Yasuteru Shigeta [2], Eriko Nango [5,7] ✉ & Takafumi Ueno [1,8] ✉

Determining short-lived intermediate structures in chemical reactions is challenging. Although ultrafast spectroscopic methods can detect the formation of transient intermediates, real-space structures cannot be determined directly from such studies. Time-resolved serial femtosecond crystallography (TR-SFX) has recently proven to be a powerful method for capturing molecular changes in proteins on femtosecond timescales. However, the methodology has been mostly applied to natural proteins/enzymes and limited to reactions promoted by synthetic molecules due to structure determination challenges. This work demonstrates the applicability of TR-SFX for investigations of chemical reaction mechanisms of synthetic metal complexes. We fix a light-induced CO-releasing $Mn(CO)_3$ reaction center in porous hen egg white lysozyme (HEWL) microcrystals. By controlling light exposure and time, we capture the real-time formation of Mn-carbonyl intermediates during the CO release reaction. The asymmetric protein environment is found to influence the order of CO release. The experimentally-observed reaction path agrees with quantum mechanical calculations. Therefore, our demonstration offers a new approach to visualize atomic-level reactions of small molecules using TR-SFX with real-space structure determination. This advance holds the potential to facilitate design of artificial metalloenzymes with precise mechanisms, empowering design, control and development of innovative reactions.

Visualizing real-space structural changes in a chemical reaction is an important area of research in fundamental and applied chemistry[1]. To explore the mechanism of a reaction, it is necessary to study the intermediate states involved in the reaction in detail. If the intermediates are transient or short-lived, it becomes a challenging task to explore the reaction in detail, particularly real-space structures[2]. Transient spectroscopic methods are usually applied to study such reaction dynamics occurring within an ultrafast

---

A full list of affiliations appears at the end of the paper. ✉e-mail: basudev@bio.titech.ac.jp; mshoji@ccs.tsukuba.ac.jp; eriko.nango.c4@tohoku.ac.jp; tueno@bio.titech.ac.jp

timescale[3,4]. Recently, femtosecond time-resolved X-ray liquidography with X-ray free-electron lasers (XFEL) was used to track atomic motions of a metal complex during bond formation reactions in solution[5–7]. While these methods allow us to characterize an electronic state and/ or a local chemical state of an atom, structural information such as the three-dimensional coordinates of each atom is limited. Therefore, it is of great interest to have a methodology for monitoring the real-space structural changes during instantaneous formation of intermediates involved in breakage and formation of chemical bonds.

Time-resolved serial femtosecond crystallography (TR-SFX) using XFEL has gained significant attention due to its ability to capture molecular snapshots of dynamic changes in protein structures in short time intervals approaching the femtosecond level[8–11]. For example, ion pumping processes in microbial rhodopsins including bacteriorhodopsin[8,11], a sodium pump[12], KR2, and a chloride pump[13], NM-R3, have been filmed over a wide time range from femtoseconds to milliseconds[8,13,14]. Similarly, an initial intermediate of NO (nitric oxide) binding to a heme protein, cytochrome P450nor (P450nor) has been investigated using this method in combination with a photo-labile caged compound[9]. Such examples suggest that TR-SFX is a powerful tool for three-dimensional structure determination of intermediates generated during a reaction occurring at ambient temperature. So far, the methodology has only been applied to biomacromolecules.

Small molecule crystals of metal complexes and organic compounds diffract to high spatial resolution in general, but the structure determination is challenging in SFX because fewer diffraction spots are generated by small molecule crystals relative to biomacromolecule crystals. Recently, a new technique known as small molecule SFX (smSFX) has been reported for determination of material structures[15,16].

This method enables structure determination by indexing and integration based on a unit cell estimated from a high-resolution powder pattern generated from SFX data.

To date, there have been no reports of time-resolved smSFX being used to observe inorganic or organic reactions of synthetic molecules, except one with a dynamic metal-organic framework (MOF) structure[17]. In another aspect, most small molecules crystallize with low solvent content and are tightly packed. Because of a limited reaction space, tight packing may interfere with the function or the reaction. In contrast, macromolecule crystals are less tightly packed. To overcome such problems and study synthetic molecule reactions by the TR-SFX method, we developed a strategy involving immobilization of the reaction center into a confined protein crystalline scaffold to permit visualization of the reaction dynamics by TR-SFX over a wide timescale. Such methodology is expected to be very useful for determination of three-dimensional structures of intermediates generated by chemical bond breakage or formation occurring at the reaction center in real-time.

To investigate this concept, we selected hen egg white lysozyme (HEWL) as a model protein template for the study. When lysozyme crystallizes, it forms porous structures with diameters ranging from 1.8 to 2.6 nm (Fig. 1a, b)[18]. The His15 located within the cavity can be utilized to immobilize metals on the pore surface (Fig. 1a). Previously, various synthetic metal complexes such as Ru(arene), cisplatin, and Mn/Ru/Re(CO)₃, have been incorporated into the porous lysozyme crystal through His15 for structure determination, observing catalytic reactions, and evaluating drug binding properties[19–21]. However, there have been no investigations of structural changes occurring during the reactions promoted by the immobilized metal centers due to a lack of suitable methods.

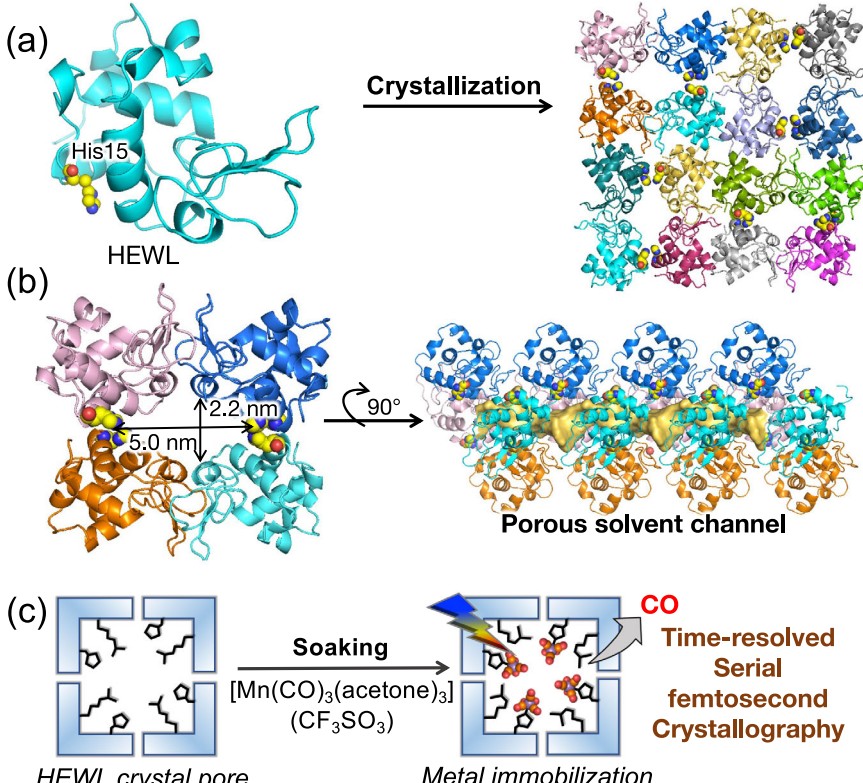

**Fig. 1 | Porous HEWL (Hen Egg White Lysozyme) crystal with a schematic representation of the research focus. a** HEWL monomer structure (PDB: 193L) and lattice packing in a tetragonal crystal displaying characteristic pores. His15, commonly known for metal binding, is depicted in a yellow ball and stick model. **b** Expanded view of a crystal pore with side view showing the channel volume highlighted in yellow. **c** Schematic representation showing the immobilization of a light-sensitive Mn(CO)₃ reaction center into the porous HEWL solvent channel and Time-resolved structure determination by Serial femtosecond crystallography after photoexcitation. Protein structures were prepared using Pymol[71] and the solvent channel was calculated using HOLLOW program[72].

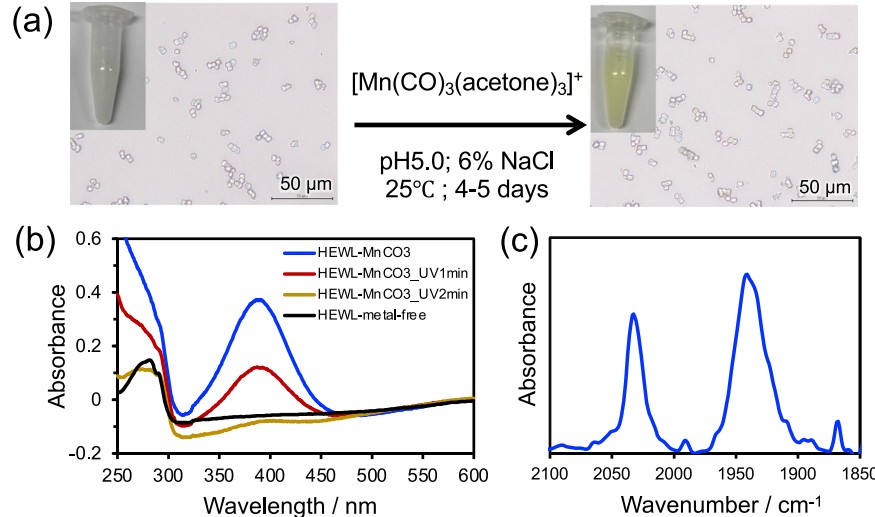

**Fig. 2 | Preparation and characterization of HEWL-Mn(CO)₃(wat)₂ microcrystals.** (**a**) Immobilization of a Mn(CO)₃ reaction center in the HEWL microcrystals. **b** UV-visible spectra of the crystal suspension for HEWL_metal-free (black) and HEWL-Mn(CO)₃ (blue) in 0.1 M acetate buffer pH5.0 (6% NaCl). The spectra of HEWL-Mn(CO)₃ microcrystals suspension after UV irradiation of 1 min and 2 min are shown in brown and yellow respectively. **c** ATR-IR spectrum of HEWL-Mn(CO)₃ microcrystals showing the CO stretching in Mn(CO)₃ reaction center. Spectra were measured after washing the crystals with 0.1 M acetate buffer (pH5.0) containing 6% NaCl.

The objective of this work was to immobilize a light-sensitive synthetic Mn(CO)₃ reaction center into porous HEWL microcrystals (<20 μm) and determine if the real-time structural changes in the metal center during the CO release reaction could be characterized by TR-SFX. Quantum mechanics/molecular mechanics (QM/MM) methods were employed to validate the experimentally observed structures[22,23]. After CO release, the Mn center will typically undergo dimerization, aerial oxidation or precipitation in solution and such results complicate mechanistic investigations[24,25]. Thus, isolation of Mn reaction center in the restricted protein environment was expected to enable a detailed experimental analysis of the intermediates generated during progression of the reaction.

Here, we show the study of a light-induced CO release reaction occurring within a crystalline protein environment using the TR-SFX method (Fig. 1c). [Mn(CO)₃(acetone)₃](CF₃SO₃) immobilized HEWL microcrystals were used for the study. By controlling the time delay and the light dose, we successfully captured the structures of key intermediates formed in real time after photoexcitation. Quantum mechanical calculations were found to agree with the experimentally determined intermediate structures. Our experiments also revealed the role of the protein environment in the light reaction. Therefore, we have established that using protein crystal as a matrix, reactions of synthetic metal complex can be studied with determination of intermediate structures.

## Results

### Preparation of HEWL microcrystals and immobilization of a Mn(CO)₃ reaction center

Small crystals (<20 μm) are better for achieving maximum photoconversion due to the limitation of light penetration into large single crystals (>200 μm)[26]. In addition, serial crystallography measures a single diffraction pattern per crystal under most experimental conditions, and therefore, a large number of microcrystals (10⁸ crystals/ml) with a typical size of about 5–20 μm are required for the X-ray diffraction measurement[27]. The HEWL crystal has tetragonal, orthorhombic, and monoclinic isomorphic forms[18]. Although the orthorhombic crystal has the largest pore (2.6 nm), we chose the tetragonal form which has a pore size of 2.2 nm for the study because it gives a very high yield of crystallization with uniform size and can be prepared in bulk scale in

less than 10 min (Fig. 1a). We prepared tetragonal HEWL microcrystals in a batch method by mixing a protein solution in 100 mM acetate buffer (pH 3.0) with an equal volume of precipitant solution containing 8% PEG6000 and 28% NaCl at 17 °C[28]. Although the size of microcrystals can be tuned by varying the temperature, we prepared crystals with an average size range of 5–10 μm for better light penetration. The harvested crystals were washed with acetate buffer (pH 5.0) containing 6% NaCl and soaked in [Mn(CO)₃(acetone)₃](CF₃SO₃) for 4–5 days in dark (Fig. 2a). For analytical characterization, the HEWL-Mn(CO)₃ microcrystals were washed with acetate buffer (pH 5.0) containing 6% NaCl to remove unbound metals. The absorption spectrum of the HEWL-Mn(CO)₃ microcrystals dispersed in acetate buffer (pH 5.0) containing 6% NaCl includes the characteristic metal-to-ligand charge transfer band at 395 nm which indicates the presence of the Mn(CO)₃ unit in the crystal (Fig. 2b). The ATR-IR spectrum of the microcrystals showed the characteristic CO stretching frequencies at 2030 cm⁻¹ and 1938 cm⁻¹, which indicate the facial arrangement of the CO ligand in Mn(CO)₃ (Fig. 2c)[20,29].

### Structure determination of HEWL-Mn(CO)₃(wat)₂ under darkness by XFEL

Radiation damage during X-ray diffraction is a major problem in many metal complexes including Mn and Fe[30–32]. To reduce the possibility of the problem occurring in the HEWL-Mn(CO)₃(wat)₂ structure, we employed serial crystallography which uses an ultra-short femtosecond pulse (<10 fs) of X-rays from the free-electron laser (XFEL) to obtain diffraction before the structure becomes damaged (Fig. 3a)[33,34]. A single diffraction pattern was collected per crystal at room temperature (RT) during the flow of HEWL-Mn(CO)₃(wat)₂ microcrystals through a nozzle (2.4 μl/min) and a minimum of 15,000 indexable diffraction patterns were collected per dataset for structure analysis[26]. Selected crystallographic data and refinement statistics are shown in Supplementary Table 1A. The structure was refined to 1.6 Å. The Mn(CO)₃(wat)₂ moiety coordinates to the His15 residue in the solvent channel (Fig. 3a–c). The coordination structure of Mn(CO)₃(wat)₂ at RT was found to be quite similar to the structure determined from data collected using single crystals at 100 K (Supplementary Figs. 1, 2 in SI) and also consistent with the previous report[20,35]. This suggests that the HEWL-Mn(CO)₃(wat)₂

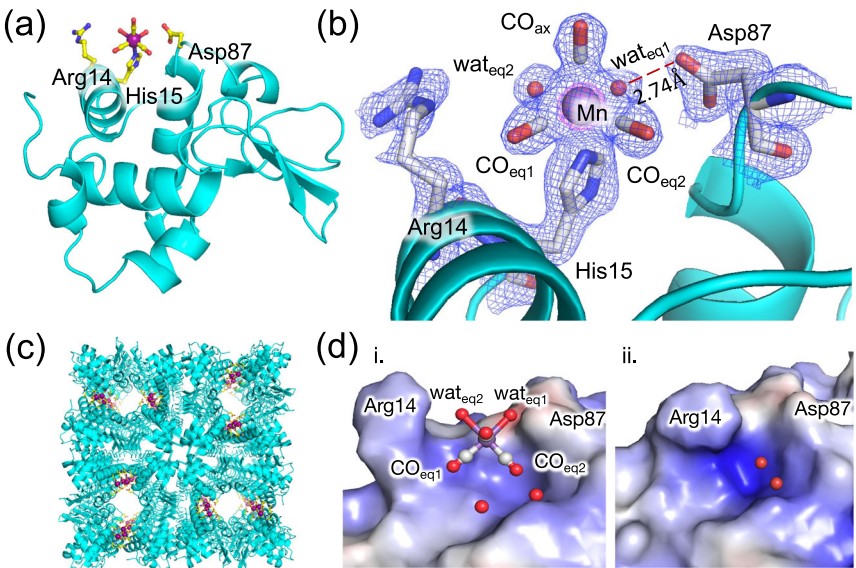

**Fig. 3 | Structure determination of HEWL·Mn(CO)$_3$(wat)$_2$ by SFX. a** Structure of HEWL·Mn(CO)$_3$(wat)$_2$ under darkness showing the position of Mn-binding and surrounding side chains. **b** Enlarged view of Mn(CO)$_3$(wat)$_2$ coordination structure at His15 showing the 2$F_o$-$F_c$ electron density map at 1σ and 7σ are shown in blue and magenta mesh, respectively. **c** Solvent channel of the tetragonal HEWL·Mn(CO)$_3$(wat)$_2$ crystal showing the positions of Mn(CO)$_3$(wat)$_2$. Mn atoms are shown in purple spheres. **d** Molecular surface models depicting the cavity in which Mn(CO)$_3$(wat)$_2$ binds (i) and the same cavity in metal-free HEWL (ii). The asymmetric protein environment makes three CO ligands inequivalent. CO$_{eq2}$ inside the cavity has the least solvent accessibility while CO$_{ax}$ exposed more to the bulk solvent. The surface models are prepared in Pymol using the APBS plugin.

structure does not have disorder at a higher temperature (RT). To test any possible CO release reaction in dark, we measured an additional set of HEWL·Mn(CO)$_3$(wat)$_2$ structures over intervals longer than 24 h, and no structural changes were observed (Supplementary Fig. 3 in SI). This excludes the possibility of any reactions occurring in the dark during the measurement period. A metal-free apo-HEWL structure was also determined as a control structure under the same conditions at RT (Supplementary Fig. 4 in SI) and the result indicated that Mn(CO)$_3$(wat)$_2$ coordination does not affect the overall protein structure because the RMSD (root mean square deviation) of the C$_\alpha$ atoms between the two structures is only 0.09 Å. It was found that during the coordination of Mn(CO)$_3$(wat)$_2$ at His15, the surrounding Arg14 moves away, possibly due to electrostatic repulsion occurring between two positive charges (Supplementary Fig. 5 in SI). The Mn center forms a slightly distorted octahedral structure at His15 with three CO and two water ligands. The CO ligands are oriented in a facial arrangement with Mn-CO distances of 1.94 Å. The distances of Mn-O$_{wat1}$, Mn-O$_{wat2}$, and Mn-N$^\varepsilon$His15 are 2.15 Å, 2.33 Å, and 2.26 Å, respectively (Supplementary Table 2). The observed bond distances are consistent with those of typical Mn-carbonyl complexes reported previously (Supplementary Fig. 2)[20,24,35]. Among the two equatorial water molecules, wat$_{eq1}$ was found to have weak interaction with Asp87 with an O(wat$_{eq1}$)-O$^\delta$(Asp87) distance of 2.74 Å (Fig. 3b). Further analysis of the surface model revealed that the unique protein environment provides three CO ligands in the asymmetric Mn(CO)$_3$(wat)$_2$ coordination structure (Fig. 3d). The CO$_{ax}$ molecule was found to be exposed to the solvent while CO$_{eq2}$ is buried within the protein cavity (Fig. 3d-i). Such an asymmetric protein environment is expected to influence the order of the CO release reaction.

**Light-induced CO release reaction with time-resolved structures**
The light reaction occurring at the Mn(CO)$_3$(wat)$_2$ center in the microcrystals was achieved by irradiating the complex with UV light of 365 nm from a nanosecond pump laser (Fig. 4a). The structures were determined at various intervals (termed as delay time, $\Delta t$) after the light excitation (6 ns) from a nanosecond pump laser (Fig. 4b) and

compared with structures determined under darkness. Both the delay time ($\Delta t$) and light dose were varied to capture the intermediates of the CO release reaction. Structural changes were analyzed in a 2$F_o$-$F_c$ map (Fig. 4c) and in an assessment of difference electron density features ($F_{light}$-$F_{dark}$) (Fig. 4d, e). To evaluate the light contamination, difference maps against interleaved dark structure and negative delay measurement were also considered (Supplementary Fig. 6).

Under darkness, the HEWL·Mn(CO)$_3$(wat)$_2$ structure has a nearly octahedral Mn coordination structure with three CO and two water ligands (Fig. 4c-i, Supplementary Table 2). The structures determined at 10 ns, 100 ns, and 1 μs after light excitation from the 20 μJ pump laser indicate a visible decrease in the 2$F_o$-$F_c$ map density of the CO$_{ax}$ ligand (Fig. 4c-ii-iv). The corresponding $F_{light}$-$F_{dark}$ difference density maps (Fig. 4d-i-iii) indicate a negative density feature in the axial position. This suggests that the Mn-CO$_{ax}$ bond exhibited greater susceptibility to light and thereby initiating the release of CO$_{ax}$ during this phase of the reaction.

At 10 ns delay, the negative density features in the difference map are mostly visible near CO$_{ax}$, suggesting that CO release is initiated at the axial position (Fig. 4d-i). It is worth mentioning that the process of release of CO$_{ax}$ and water exchange might occur simultaneously during the reaction and thus, it is challenging to model the exchange of a water molecule for the residual CO$_{ax}$ ligand during the release in the 2$F_o$-$F_c$ map (Fig. 4c-ii, Supplementary Fig. 7). Considering the occupancy and B-factor of the C and O atoms of the CO$_{ax}$ ligand, we assigned a water molecule to the axial position of the structure observed 10 ns after light excitation (Fig. 4c-ii). The same feature was observed for the structure after a delay of 100 ns (Fig. 4c-iii). The moment when the CO ligand is about to dissociate, and the water molecule is about to coordinate may not constitute a stable intermediate or transition state. Therefore, TR-SFX is unlikely to capture this particular moment. The release of the CO$_{ax}$ ligand suggests that the formation of a biscarbonyl, Mn(CO$_{eq}$)$_2$ species is one of the key intermediates of the CO release reaction which was visualized during real-time observations. The result is consistent with the formation of the previously-identified Mn-biscarbonyl intermediate in theoretical

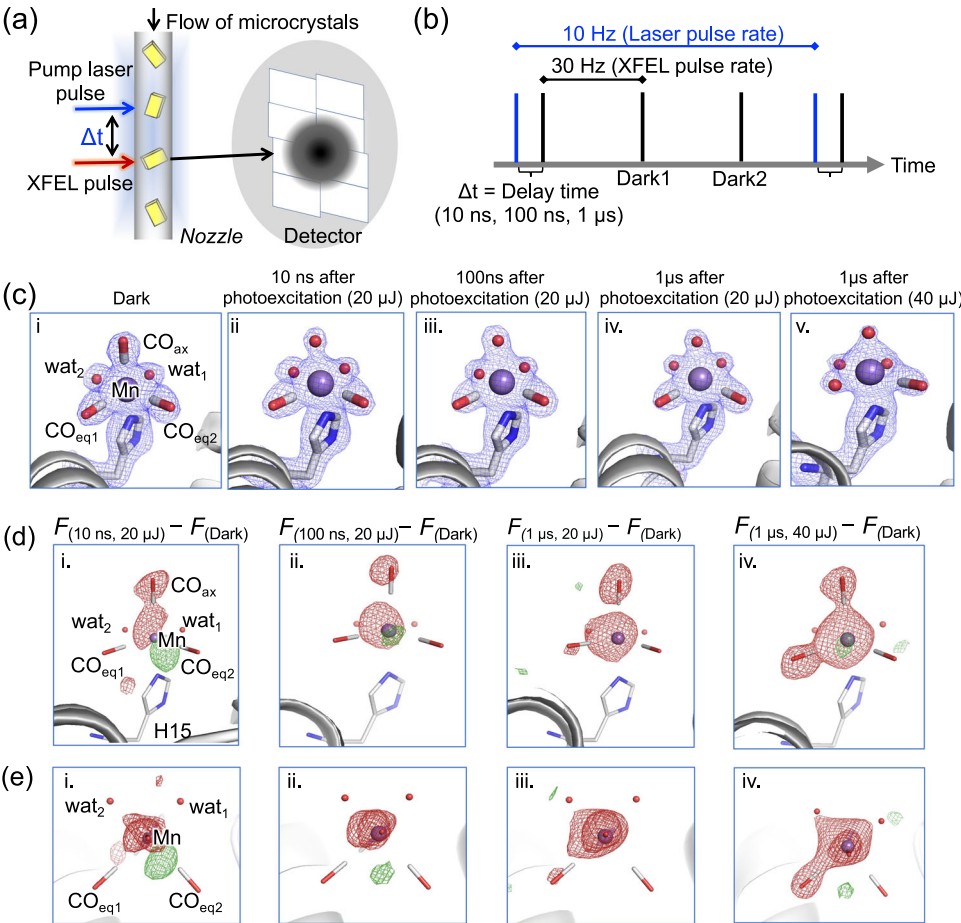

**Fig. 4 | Observation of light-induced CO release reaction and structure analysis.**
**a** Schematic representation showing the experimental setup for TR-SFX measurement. **b** A typical setup showing the alignment of XFEL pulse (repetition: 30 Hz) and pump laser pulse (repetition: 10 Hz). Time difference between the pump laser pulse and the XFEL pulse is considered as the delay time ($\Delta t$) after photoexcitation. **c** Changes in the $2F_o$-$F_c$ maps (at $1\sigma$, blue) during the progress of CO release reaction inside the HEWL microcrystals at various time delay after the 6 ns of photoexcitation. Values in parenthesis represents pump laser power. **d** The difference $F_{light}$-$F_{dark}$ map (at $\pm 3.2\sigma$, green/red) with respect to structure under darkness showing the ligand release during the reaction. $F$ represents the observed structure factor amplitude in the difference map. Complete darkness model was used for phases when calculating these difference maps. **e** View of the difference $F_{light}$-$F_{dark}$ map (at $\pm 3.2\sigma$, green/red) from the direction of $CO_{ax}$ corresponding to the images as shown in (**d**).

calculations and transient absorption/IR spectroscopic studies on related synthetic $Mn(CO)_3$ complexes[24,25,36]. The stability of such a Mn-biscarbonyl species is known to depend on the associated ligands and interaction with oxygen. A positive density feature was observed near Mn which might indicate movement of Mn due to changes in coordination structure occurring in the $CO_{ax}$ release (Fig. 4d–i, e–i). The overlap structures showed the movement of 0.098 Å (Supplementary Fig. 11d). The density is reduced with longer delay times.

When the delay time was increased from 100 ns to 1 μs, a weak negative density features appear at $CO_{eq1}$ (Fig. 4d-iii). This suggests that release of $CO_{eq1}$ begins at this stage as a second step in the process of CO release. Pronounced negative density features on the Mn center appeared after the 1st CO release. This could be due to two reasons: (i) The Mn(I) is known to get oxidized after the CO release[24,25] and that changes the electronic environment at the Mn ion, and (ii) parallel reactions also occur, which promote release of the whole Mn unit to some extent (Supplementary Fig. 17). This supports the observed decrease in Mn occupancies (Supplementary Table 3, Supplementary Fig. 9Ad, Bd). As shown in Supplementary Fig. 17, the energy barrier to release the whole Mn-unit (release of Imidazole (His)-HEWL) is close compared to the assigned intermediate with one CO release. A negative density feature was not observed at $CO_{eq2}$. In the comparison with the metal-free HEWL structure, a water molecule is positioned in place

of $CO_{eq2}$. This may explain why a negative feature is not observed (Supplementary Fig. 4).

When the light dose was increased from 20 μJ to 40 μJ while retaining the same delay (1 μs), a significant change in the $2F_o$-$F_c$ map was observed with a very weak density at the axial ligand (Fig. 4c-v). Two unequal densities for the equatorial CO ligands in the $2F_o$-$F_c$ map indicate majority or complete release of $CO_{eq1}$ at this stage of the reaction (Fig. 4c-v and Supplementary Fig. 8). As observed for initial $CO_{ax}$ release (Fig. 4c-ii), it is challenging to distinguish the $CO_{eq1}$ ligand from a single exchanging water molecule or the residual CO ligand because the process occurs simultaneously. Considering the B-factor, occupancy, and shape of the electron density maps, a water molecule was assigned to the $eq_1$ position (Fig. 4c-v). The negative density feature in the corresponding $F_{light}$-$F_{dark}$ difference map (Fig. 4d-iv, e-iv) also supports the release of $CO_{eq1}$ and possible formation of a Mn-$CO_{eq2}$ species. Therefore, this represents a signature of the real-time formation of another intermediate of the CO release reaction. The CO release was previously found to be associated with solvent exchange[24]. The asymmetric protein environment shown in Fig. 3d provides variable solvent accessibility to three CO ligands and this is reflected in their order of release. The intermediate observed after release of two CO ligands was also found to be energetically favorable as seen in the QM/MM results described in the next section.

Notably, the water molecule that was exchanged with the $CO_{ax}$ ligand at the initial stage of the reaction (Fig. 4c-ii-iv) appears to be quite weak in the $2F_o$-$F_c$ map possibly due to disorder (Fig. 4c-v). Since this position is highly exposed to the solvent channel (Fig. 3d), dynamic exchange with the bulk solvent could be a reason for the weak density. In addition, the Mn center is expected to be more dynamic due to changes in the coordination structure occurring during CO release.

We performed a power titration at 10 ns and 1 μs delay at which the laser intensity was varied. It was found that the negative density features increased with increasing laser intensity, suggesting that the laser intensity significantly influenced the reaction (Supplementary Table 4 and Supplementary Fig. 9). To estimate the apparent occupancy of released CO or Mn, we used a similar omit map based method reported by Barends et al. in which the difference in omit map density between the dark state and light state represents the amount of released CO or Mn[37]. The apparent occupancy of released CO was found to increase with increasing the incident laser energy density (Supplementary Fig. 9Ad, Bd). A nonlinear increase of occupancy (based on omit map density) after 20 μJ indicates the possibility of damage by higher laser intensity[37,38]. The occupancy of $CO_{eq2}$, located below the Mn, suggests the involvement of a free water molecule that typically occupies the position of $CO_{eq2}$ in metal-free HEWL or after the complete release of Mn-carbonyl (Supplementary Fig. 11a–c).

To observe later stages of the CO release reaction, we measured the structures at longer delay times of 1 ms and 17 ms (Supplementary Fig. 10). At longer delay times, a very weak negative density feature towards $CO_{eq2}$ (close to Mn and atom C of $CO_{eq2}$) was observed (Supplementary Fig. 10b). This might be an indication of $CO_{eq2}$ release. The absence of a negative density feature on the atom O of $CO_{eq2}$ agrees with the fact that a water molecule takes that position in metal-free HEWL (Supplementary Fig. 11a). It is to be noted that during later stages of the reaction, there is a possibility of simultaneous formation of multiple reaction intermediates. In addition, the possibility of re-bounding the released Mn cannot be excluded because the crystals were coated with high-viscosity grease, and the released Mn might not escape out of the crystal. Therefore, the structure observed at a longer delay time (1 ms or 17 ms) represents an average structure (Supplementary Fig. 10b, c, e, f), and the observed difference density features are not in perfectly descending order when compared with structures of 10 ns, 100 ns or 1 μs delay time (Fig. 4d). Considering all, we can say that our method is more suitable to study initial stages of the CO release reaction.

To observe the structure after complete release of CO from both protein and bulk solvent, we irradiated light from a normal UV lamp (365 nm) on the single crystal of HEWL-Mn(CO)$_3$(wat)$_2$ for over 20 min. The yellow color of the crystal disappeared and the structure indicated complete release of Mn(CO)$_3$(wat)$_2$. This structure is similar to the metal-free HEWL structure (Supplementary Fig. 11c). After the release of CO was complete, the Mn-aqua species was expected to be unstable and to be released immediately from the protein. Previous reports indicated that the Mn$^I$(CO)$_3$ moiety after the CO release gets oxidized to Mn$^{II}$(CO)$_3$(wat) which is less stable and subsequently trigger the release of other CO ligands, reactions with oxygen to form MnO$_x$[24,25,39]. Notably, a water molecule was appeared in place of $CO_{eq2}$.

During Mn(CO)$_3$(wat)$_2$ coordination, Arg14 moves away from His15 (the original position in metal-free HEWL). This could be due to electrostatic repulsion from the positively charged metal center (Supplementary Figs. 5, 11a, b, c). After complete CO release, the conformation of Arg14 significantly changes due to relaxation of electrostatic repulsion (Supplementary Fig. 11c). During TR-SFX measurements, we also visualized the intermediates of conformational change (Supplementary Fig. 11d). Asp87 appears to adopt multiple conformations during the light reaction (Supplementary Fig. 11d). Asp87 has a weak interaction with wat$_{eq2}$, which was directly coordinated to Mn and thus, directly participates in the CO release reaction. (Supplementary Fig. 5) All such observations suggest that the protein environment cooperatively induces the light-triggered CO release reaction.

## QM/MM analysis on the photoinduced CO release reaction path

We performed QM/MM calculations[40] to elucidate the detailed reaction processes for the HEWL-Mn(CO)$_3$(wat)$_2$ photoreaction. Reaction energy profiles and UV-visible absorption spectra measured during the CO-wat exchange were evaluated using a QM/MM model (Fig. 5). Using the SFX structure of HEWL-Mn(CO)$_3$(wat)$_2$ measured in the dark, a QM/MM model composed of the HEWL monomer in a water droplet was constructed. Detailed computational procedures for setting up the QM/MM model are described in the Methods section and Supplementary Information. The entire QM/MM system is illustrated in Supplementary Fig. 12. The optimized QM/MM structure of HEWL-Mn(CO)$_3$(wat)$_2$ (**1**) remains essentially identical in the original SFX structure, and the RMSD for the heavy atoms in the QM region is 0.228 Å (Supplementary Fig. 13), where the Mn-ligand distances are only slightly shortened to 2.13 Å (Mn-O(wat$_{eq}$)), 1.80 Å (Mn-C(CO$_{eq}$)) and 2.12 Å (Mn-N$^\epsilon$(His15)) in **1** (Supplementary Table 5). UV-visible spectrum of **1** calculated at the QM/MM level of theory has a characteristic absorption with a peak at 394 nm (black curve in Fig. 5a), which is in good agreement with the experimental 395 nm result. We checked that the used QM/MM level is converged enough for the calculation of the absorption spectra against the improvement of the basis set and QM region (Supplementary Fig. 14 and Supplementary Table 7).

All reaction pathways for CO-wat exchange reactions were searched by using the nudged elastic band (NEB)[41], which can calculate minimum energy pathways (Supplementary Fig. 15). By using a less computational cost QM/MM level of theory, all of the energy profiles along the reaction steps are determined. After following the lowest-energy processes among all possible ligand-water exchange reactions (Supplementary Table 8), the most energetically preferable reaction was found to take the following steps: (1) CO$_{ax}$-wat exchange (**1 → 2**), (2) CO$_{eq1}$-wat exchange (**2 → 3**), and (3) His15 imidazole (Im)-wat exchange (**3 → 4**). The atomic coordinates of all the intermediates are shown in Supplementary Table 9. It should be noted that the activation energy of wat$_{eq2}$-wat exchange is lowest in energy with $\Delta E = 3.2$ kcal mol$^{-1}$. However, as this reaction doesn't change the overall ligand coordination after the reaction, this wat-wat exchange reaction is excluded from the actual ligand exchange reaction.

The energy barriers in the first excited state (S$_1$ state) were calculated to be $\Delta E^*(S_1) = 6.0$, 10.2, and 8.3 kcal mol$^{-1}$ for the first, the second, and third ligand-water exchange reactions, respectively. It is stressed here that they are lower compared to the energy barriers in the ground state ($\Delta E^*(S_0) = 20.0$, 24.1, and 19.9 kcal mol$^{-1}$ (Fig. 5b)). As the ligand-water exchange reactions proceed, the energy of the first excited state remains unchanged ($\Delta E = 0.2$ (**1 → 2**), −5.1 (**2 → 3**) and 1.3 (**3 → 4**) kcal mol$^{-1}$), while the state of the ground state becomes much higher in energy ($\Delta E = 16.3$ (**1 → 2**), 17.5 (**2 → 3**) and 4.0 (**3 → 4**) kcal mol$^{-1}$). These features in the energy profile, i.e., low activation barrier and very low endothermic reaction steps, explain the high reactivity of the Mn(CO)$_3$ center in the first excited state. Calculated UV-visible spectra for these intermediate states (**1–4**) in Fig. 5a indicate that the absorption peak moves to a longer wavelength and the absorbance decreases ($\lambda_{max} = 483$ (**2**), 548 (**3**) and 544 nm (**4**)) as the reaction proceeds. This change in the UV-visible absorption is consistent with the color change of HEWL-Mn(CO)$_3$ observed upon the UV light irradiation. Molecular orbitals contribute to the first excited state as shown in Supplementary Fig. 16. The highest occupied molecular orbitals (HOMOs) are mainly distributed on the Mn d orbitals, and the lowest unoccupied molecular orbitals (LUMOs) are more delocalized to the CO and water ligands with antibonding character. These MO

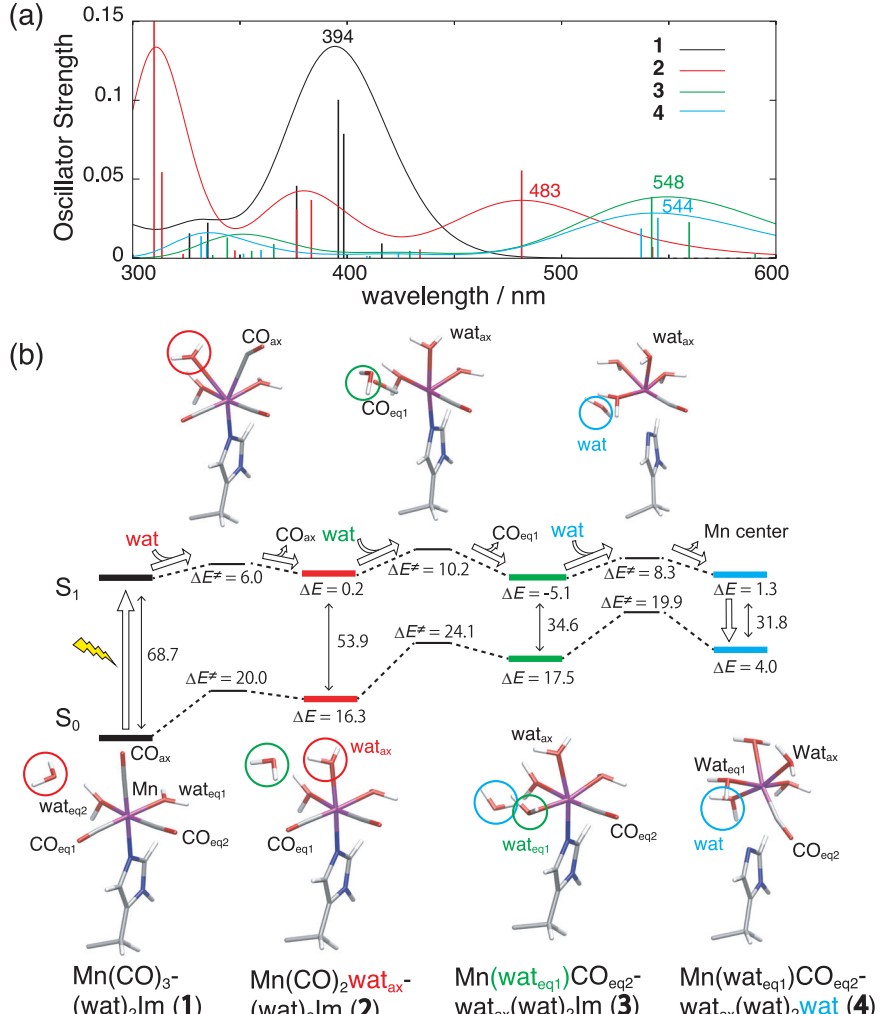

**Fig. 5 | QM/MM results of HEWL-Mn(CO)$_3$(wat)$_2$ for the CO-wat exchange reaction. a** Calculated UV-visible spectra for the intermediate states (**1**–**4**). **b** Energy diagram for each of the CO-Wat exchange reactions. Optimized molecular structures in the ground and first excited states are depicted below and above the energy diagram. Only the atoms of the Mn center are shown for clarity. Relative energies are given in kcal mol$^{-1}$. Entire QM/MM system and the explicit QM regions which changed on the calculated states are illustrated in Supplementary Figs. 12, 14. Reaction barriers for other higher-activation pathways are summarized in Supplementary Figs. 15, 17, and Supplementary Table 8.

transitions also support the enhancement of the ligand dissociation via the excited states.

This QM/MM expected reaction ($1 \rightarrow 2 \rightarrow 3 \rightarrow 4$) is consistent with the reaction observed in the TR-SFX measurements. From a structural point of view, elongations of the Mn-ligand bond length upon the CO release are also consistent (Supplementary Table 5), and the Mn center of the structure in complete dark, 10 ns and 1 μs delay becomes closest to one of **1**, **2**, and **3** of QM/MM optimized structures based on the values of root-mean-square deviation (RMSD) (Supplementary Table 6). We note that reactions with energy barrier of 6.0 kcal mol$^{-1}$ and 9.2 kcal mol$^{-1}$ corresponds to a reaction step with a reaction rate of 3.7 ns and 0.8 μs, respectively. The energy barriers of 6.0 kcal mol$^{-1}$ and 10.2 kcal mol$^{-1}$ evaluated by using QM/MM correspond to the reactions directly observed by TR-SFX measurements in 10 ns and 1 μs delays, respectively. A more detailed discussion about the reaction rate is found in the Supporting Information.

Product Mn aqua complexes are expected to be more unstable and tend to dimerize, aggregate or decompose as reported previously[24]. Unfortunately, in our TR-SFX experiments, we could not precisely determine the structural changes at the later stage of the reaction (1 ms delay or longer) due to the potential presence of multiple intermediates. Therefore, a direct observation of the release of

the Mn-aqua species is expected to be very difficult to follow experimentally. The present QM/MM calculation predicts that the release of the Mn(CO)-unit ($3 \rightarrow 4$) is faster ($\Delta E^{\neq}(S_1) = 8.3$ kcal·mol$^{-1}$) compared to the third CO$_{eq2}$-Wat exchange with a higher activation barrier ($\Delta E^{\neq}(S_1) = 32.3$ kcal·mol$^{-1}$).

The reaction pathway which includes earlier release of the Mn center is the second lowest in energy for the activation barrier after CO$_{ax}$-wat exchange, namely, via (1) CO$_{ax}$-wat exchange ($1 \rightarrow 2$) and (2) Im-wat exchange ($2 \rightarrow 5$) (Supplementary Fig. 17 and Supplementary Table 8). The product along this reaction is MnCO$_{eq1}$CO$_{eq2}$wat$_{ax}$(wat)$_3$ (**5**). The activation barrier of the second reaction in the $S_1$ state was calculated to be $\Delta E^{\neq}(S_1) = 14.7$ kcal·mol$^{-1}$ ($2 \rightarrow 5$). This pathway corresponds to the early change of Mn occupancy observed in the TR-SFX structure after 10 ns between 20 μJ and 40 μJ photolysis.

Based on the present QM/MM calculations, we determined that the increase of the activation barrier of CO-wat exchange reaction is caused by blocking access of ligands to the external solvent. When a ligand is surrounded by a protein environment, the energy barrier of the ligand-water exchange reaction becomes higher, which must form an unstable open-coordination state (five-coordinated state) compared to a more stable seven-coordinated state (such as the transient state occurring during $1 \rightarrow 2$ in Fig. 5b). CO$_{ax}$ is the CO molecule which

is most exposed to solvent among the three Mn-coordinated CO molecules, and $CO_{eq2}$ has the most exposure to surrounding amino acids, which are the hydrophobic side chains of Ala11 and Ile88. $CO_{eq1}$ is partially covered by the side chain of Arg14 (Fig. 3d-ii). The different reactivity of the CO-wat exchange reaction of HEWL-Mn(CO)$_3$ is understood to be due to the asymmetrical protein environment around the Mn center.

## Discussion

The present work demonstrated the capture of real-time formation of intermediates in a typical CO release reaction within the porous HEWL microcrystal using the TR-SFX method. The intermediates formed in a CO release reaction in solution involve oxidation of Mn(I) reaction center followed by the release of other CO ligands, reactions with oxygen, and precipitation[24,25]. Thus, it is difficult to study the formation of the intermediates in solution except by transient spectroscopic studies[42]. In our study, the unique protein cavity prevents the Mn(I) reaction center from undergoing dimerization, aggregation precipitation, etc. It enables determination of real-time structural changes occurring at the Mn center at the atomic level by the TR-SFX method. Previously, femtosecond transient absorption spectroscopy was used to study the CO release reaction in a solution of synthetic Mn-tricarbonyl complex[43]. Only one of the three CO ligands was observed from the femtosecond transient absorptions followed by the replacement of the free coordination site by a solvent molecule. However, determination of the structure after the CO release was dependent on the quantum chemical calculations. In contrast, the TR-SFX method directly captured the real-time intermediate formation. The intermediates were also verified using the QM/MM method. However, if one considers any potentially non-productive pathways, it cannot be completely ruled out that other intermediates may form in small fractions. This is because the energy barriers for forming those intermediates are not very large compared to the assigned stable intermediates (Supplementary Fig. 17). Thus, the reaction path described in this paper represents the primary route for the CO release reaction.

This is significant in considering the sensitive reactions which are difficult to study in solution. The TR-SFX methodology has so far been generally limited to investigating dynamic changes in natural proteins during small molecule reactions such as NO binding in NO reductase and CO release from a CO-bounded heme protein[8,9,44]. Therefore, our studies have increased the scope of applications of TR-SFX to include studies of synthetic molecule reactions within protein crystals.

With broader applicability in the context of MOFs, crystalline sponges, etc.[17], one can consider these crystalline matrices. Although they show promise and have a wider range of applications, the inherent difficulty lies in their structure determination by SFX[16]. Structural analysis of small molecules like rhodamine, AgSePh, etc. has recently been achieved using the smSFX method[15,16]. However, if we consider the structural changes occurring during a reaction process, it might be challenging to use this method because the reactions are spatially restricted due to tighter crystal packing of small molecule crystals. In contrast, a protein crystal can be considered as a better reaction matrix because it can be functionalized easily with synthetic molecules and flexible side chains can trap intermediates without affecting the crystal lattice. Therefore, studying small molecule reactions within a protein crystal matrix can be considered as a significant advance which we demonstrate in this work using a typical CO release reaction.

Since the HEWL crystal used as a model template can immobilize various organometallic complexes in the crystal[45], their catalytic reactions can be observed directly by TR-SFX. In addition, protein crystals generally form pores with tens to hundreds of nm, which can provide appropriate chemical environments for investigating reactions[45–47]. Thus, the TR-SFX experiment of synthetic metal complexes immobilized in the HEWL crystals is the first appropriate demonstration showing the potential of our method.

The TR-SFX method identifies the intermediates based on changes in electron density maps. It is occasionally difficult to assign the exact intermediate in the structure if parallel reactions occur simultaneously at the metal center. At the initial stage of the reaction, we observed the release of $CO_{ax}$ with the formation of a Mn-bicarbonyl species. During the formation of this intermediate, the Mn occupancy remains almost same (0.9). Interestingly, we also identified the second instance of CO release in real-time (Fig. 4d-iv) and this become possible because due to the asymmetric protein environment. Such types of intermediates have not been identified previously. The formation of such intermediates is also in agreement with quantum mechanical chemical calculations. These results indicate the significance of using a protein scaffold for studying small molecule reactions.

While our method focused on describing photoinduced CO release reactions using a synthetic metal complex, the protein template with immobilized metal could serve as a versatile platform for exploring the mechanism of reactions promoted by photocatalysis[17,48,49]. In addition, a mixed-jet system can be employed to explore a broader range of reactions[50–52]. In this setup, the protein crystal can be mixed with the reaction substrates or trigger compounds to initiate the reactions without light irradiation.

In summary, we have successfully demonstrated the fixation of a typical CO-releasing light-sensitive Mn(CO)$_3$ moiety into the porous HEWL microcrystals and studied the reaction dynamics of CO release using the TR-SFX method. This has enabled us to determine the structures of the HEWL-Mn(CO)$_3$(wat)$_2$ in short time intervals during the progression of the light reaction. We captured the initial release of the axial CO ligand and determined the real-space structure of an Mn-biscarbonyl species as a reaction intermediate. We also directly visualized the involvement of protein side chains and their conformational changes occurring during the reaction. Therefore, the current results revealed the mechanism of a typical CO release reaction which was previously identified only by spectroscopic and theoretical studies. Our approach provides a simple and effective method to study structural dynamics at a reaction center in real-time and real space. The method shows potential for investigating the reaction mechanisms of synthetic bio-inspired catalysts and enzymatic systems incorporating non-biological components.

## Methods

### Materials and general methods

Hen egg white lysozyme and Mn(CO)$_5$Br was purchased from Sigma-Aldrich and used as received. Other chemicals like acetic acid, sodium acetate, NaCl, PEG6000, etc. were purchased from TCI, Wako chemicals, etc., and used as received. The UV-visible spectra were measured on a UV-2400PC UV–vis spectrometer (Shimazu). The ATR-IR measurement was done using FT-IR4200 instrument (JASCO).

### Preparation of lysozyme microcrystals

HEWL microcrystals were prepared by following previous reports with minor modifications[28]. Briefly, 20 mg/ml of HEWL in 0.1 M acetate buffer, pH3.0 was added into the equal volume of precipitant containing 28% NaCl and 8% PEG6000 in 0.1 M acetate buffer, pH3.0 at 17 °C (700 rpm). The crystals appeared within 5 min and centrifuged the tube to replace the buffer with precipitant. The process was repeated thrice to remove uncrystallized protein. The microcrystallization was found to be very sensitive to temperature. The microcrystals were stored in the refrigerator.

### Immobilization of [Mn(CO)$_3$(wat)$_3$]$^+$ into HEWL microcrystals

The Mn-carbonyl precursor complex, [Mn(CO)$_3$(acetone)$_3$](CF$_3$SO$_3$) was prepared by following a similar procedure as reported before[29]. 150 mg of Mn(CO)$_5$Br in 7 ml acetone was mixed with 1 equivalent of AgCF$_3$SO$_3$ and allowed to reflux overnight at 70 °C under argon atmosphere. The reaction was carried out with minimum exposure of

light. Then, the mixture was filtered and the supernatant was evaporated to obtain a brown oil. 25 μl of the brown oil was mixed with 0.5 ml of 0.1 M acetate buffer, pH 5.0 containing 6% NaCl and vortex. The mixture was then, syringe filtered (0.43 μm) to obtain a clear brown-yellow solution which was added into the HEWL microcrystals which was previously washed with 0.1 M acetate buffer, pH 5.0 containing 6% NaCl and allowed to soak (cyclo-mixture) for 4–5 days at RT followed by XFEL diffraction. $[Mn(CO)_3(wat)_3]^+$ is known to form in situ by aquation of $[Mn(CO)_3(acetone)_3](CF_3SO_3)$ and can react with HEWL[20,29].

### Absorption and ATR-IR spectral measurement
The metal soaked HEWL microcrystals were washed with 0.1 M acetate buffer, pH 5.0 containing 6% NaCl to remove unbounded metals. Then, the suspended microcrystals in 0.1 M acetate buffer, pH 5.0 containing 6% NaCl was used for absorption spectral measurement. Similarly, the metal-bounded HEWL crystals was placed on the IR probe for ATR-IR measurement.

### XFEL measurement
XFEL experiments were performed at BL2 of SACLA[53,54]. The XFEL was operated with a repetition rate of 30 Hz, a photon energy of 10 keV, a pulse duration of <10 fs, and a focal spot of 1.5 μm in full width at half-maximum (FWHM). A nanosecond pump pulse with a wavelength of 365 nm was used at a repetition rate of 15 Hz or 10 Hz for the time-resolved experiments. In the case of 15 Hz, the light and dark images were collected alternately since the repetition rate of XFEL was 30 Hz. On the other hand, in the case of 10 Hz, the interleaved dark images were measured twice after a light image. 20 μJ pump pulses were focused on a beam size of 100 μm (FWHM). All dataset were collected using a MPCCD-phase III detector[55]. The flow rate of the sample was 2.4 μl/min, crystal traveling speed was 9.0 mm/s, crystal traveling distance between pulses was 300 μm and the diameter of the nozzle was 75 μm.

Preparation of grease-dispersed microcrystals, SFX data collection and data processing was done according to the previous reports[56]. Briefly, a 200 μl of microcrystal containing the metal soaking solution (~$10^8$ crystals/ml) was centrifuged (tabletop centrifuge) followed by removal of 170 μl of supernatant. The remaining microcrystals were mixed with 200 μl of super Lube nuclear grade grease (No. 42150, Synco Chemical Co.) on a plastic plate. Then, the grease-dispersed Lys-MnCO3 microcrystals were transferred into a 200 μl sample cartridge of a high-viscosity cartridge-type injector[56]. To keep the temperature maintained at 293 K, a cooling jacket of the injector was used. However, the temperature at a nozzle part of the injector was not controlled. Therefore, the sample temperature was presumed to have elevated to 299–300 K during measurement.

### XFEL data processing
A real-time data processing pipeline[57] based on Cheetah[58] and CrystFEL[59,60] guided data collection and found and wrote hit images as HDF5 files. The detector geometry was refined with geoptimiser[61]. Indexing was performed by the XGANDALF algorithm[62]. Calculations were parallelized by GNU parallel[63]. The datasets were collected over four sessions. For each session, merging strategies were optimized by testing no scaling, scaling in *process_hkl*, and scaling in *partialator*, while the per-image resolution limit by the *push-res* parameter was varied. The resulting strategies were scaling by partialator without partiality correction, push-res = 0.75 for datasets collected in 2021 November and 2.00 (2024 February) or Monte Carlo integration by process_hkl without scaling, push-res = 0.75 (2020 November) and 1.25 (2022 October). Friedel pairs were merged. Although some datasets diffracted to higher resolutions, all datasets were truncated at 1.6 Å to facilitate difference map comparison.

### Computational details
The X-ray crystal structure used for the QM/MM calculations was the HEWL structure in dark with a resolution of 1.7 Å. A monomer model

was solvated into a water droplet with a 30 Å radius and the whole system was kept in a neutral charge by replacing some water molecules with eight $Cl^-$ ions. The whole system was equilibrated via classical molecular dynamics (MD) within the Amberff99 and TIP3P force fields under fixing the coordinates of heavy atoms determined by the X-ray crystallography[64]. The equilibration process was annealing MD at 300 K for 2 ps to relax the solvent water molecules and added hydrogen atoms for the lysozyme.

After the annealing, QM/MM model was constructed. The QM region consists of a side chain of His15, Mn(I), three CO, and two water molecules (Supplementary Fig. 12). For the electronic structure description, the density functional theory (DFT) at B3LYP-D3/DZVP was resorted. The remaining part of the system was treated at the same MM level with the Amberff99 and TIP3P force field as used in the equilibration. B3LYP-D3 means hybrid exchange-correlation functional B3LYP with Grimme's D3 dispersion correction, and the basis set DZVP is double zeta plus polarized one, LANL-2DZ ECP for Mn atom and 6-31G* for other atoms. The DFT method can properly reproduce the relative stabilities and geometrical structures in small size molecules and larger size molecules such as active sites containing transition metal centers[65,66]. Geometry optimization was carried out for all the atoms within a 10-Å radius from the original Mn center. An electronic embedding scheme and hydrogen atom link were adapted for the QM-MM interface, and non-bonded interactions between QM and MM regions are explicitly computed without introducing a cutoff for all energy calculations. Reaction pathways and transition states were searched with the NEB method[67]. We used 13 and 19 images for each reaction pathway. In the NEB calculations, reactions of a ligand exchange to a solvent water molecule were investigated. The nearest solvent water molecule was included in the QM region depending on the reaction steps (Fig. 5). Energy profiles in the excited states were evaluated on the NEB optimized pathway in the ground state, which is an approximation but practical to reduce the computational demands for a high number of reaction steps. The excited states were calculated at the time dependent DFT (TDDFT) with the B3LYP-D3/DZVP. UV–Vis absorption spectra calculations were performed at TDDFT with CAMB3LYP/TZVP level, where TZVP means basis sets of LANL-2TZ for Mn atom and 6-311 + G* for other atoms[68,69]. QM region for the UV-vis spectra calculation includes the side chain of Asp87 to neutralize the total charge of the QM region in the QM/MM system (Supplementary Fig. 14). All the MD and QM/MM calculations were performed using the NWChem 6.8 program package[40]. The molecular structures shown in Fig. 5 were drawn using the VMD program[70].

### Reporting summary
Further information on research design is available in the Nature Portfolio Reporting Summary linked to this article.

## Data availability
All the data described in the paper are presented either in the main text or with supporting information. Atomic models and map coefficients were deposited to the Protein Data Bank (PDB) with accession code 8WZF, 8WZG, 8WZR, 8WZT, and 8WZV (https://www.rcsb.org/). Raw diffraction images of all the structures used in this paper were deposited to CXIDB with accession ID 221 (https://doi.org/10.11577/2203883). Trajectories for all the 16 reaction pathways calculated using the QM/MM NEB method have been deposited to the Biological Structure Model Archive (BSM-Arc) under BSM-ID BSM00067 (https://doi.org/10.51093/bsm-00067).

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

## Acknowledgements

We sincerely thank the SACLA engineering team for the XFEL measurement and SACLA HPC system for the computational environment. This work used storage resources at RIKEN R-CCS and University of Tokyo provided through the HPCI System Research Project (Project ID: hp230365). We thank Sam-Yong Park and Kenji Mizutani for the initial assessment of structure determination using single crystals. We also thank Mr. Tomoyuki Tanaka for his support in initial XFEL measurements. The XFEL experiments were performed at SACLA BL2 with the approval of the Japan Synchrotron Radiation Research Institute (JASRI) (Proposal No. 2019A8052, 2019B8029, 2019B8053, 2020A8034, 2021B8024, 2022B8005, 2023B8003, 2023B808062). This work was also supported by the Japan Society for the Promotion of Science KAKENHI Grants No. 19H05781 (E.N.) and 19H05776 (S.I.), 23H04879 (Y.S.), 20H05438 (B.M), 22H04744 (B.M), 23K04928 (B.M), 18H05421 (T. U.), 22H00347 (T.U.); the Platform Project for Supporting Drug Discovery and Life Science Research from the Japan Agency for Medical Research and Development under Grant No. JP21am0101070 (S.I.); CREST program from the Japan Science and Technology Agency Grant No. JP20338388 (Y.S.).

## Author contributions

Conceptualization of the work: B.M., E.N., and T.U. Crystallization, metal complex synthesis, and sample preparation: B.M. X-ray diffraction data processing: T.N. and F.L. Structure analysis: B.M, F.L., E.N., S.A. SACLA-XFEL measurement: B.M, F.L., S.A., T.T.P., M.K., Y.H., J.T., J.T., M.N., T.S., H.N., Y.N., A.A., X.Y. Organizing the instrumental section: E.N., S.O., J.K., K.T., R.T., S.I. Computational studies: M.S. and Y.S. Data curation: B.M., M.S., F.L., T.N., S.A., Y.S., E.N., T.U. The original draft was written by B.M., E.N., M.S., and T.U. with contributions from all authors. All the authors have reviewed and approved the manuscript.

## Competing interests

The authors declare no competing interests.

## Additional information

¹School of Life Science and Technology, Tokyo Institute of Technology, Nagatsuta-cho 4259, Midori-ku, Yokohama, Japan. ²Center for Computational
Sciences, University of Tsukuba, 1-1-1 Tennodai, Tsukuba, Ibaraki 305-8577, Japan. ³JASRI, 1-1-1, Kouto, Sayo-cho, Sayo-gun, Hyogo 679-5198, Japan. ⁴Institute
of Protein Research, Osaka University, Osaka, Japan. ⁵RIKEN SPring-8 Center, Hyogo 679-5148, Japan. ⁶Department of Cell Biology, Graduate School of
Medicine, Kyoto University, Kyoto, Japan. ⁷Tohoku University. Institute of Multidisciplinary Research for Advanced Materials, Tohoku University, Sendai, Japan.
⁸Research Center for Autonomous Systems Materiology (ASMat), Institute of Innovative Research, Tokyo Institute of Technology, Nagatsuta-cho 4259, Midori-
ku, Yokohama, Japan. ✉e-mail: basudev@bio.titech.ac.jp; mshoji@ccs.tsukuba.ac.jp; eriko.nango.c4@tohoku.ac.jp; tueno@bio.titech.ac.jp

