## [Peer Review File · Nature Communications]

Real-time observation of a metal complex-driven reaction intermediate using a porous protein crystal and serial femtosecond crystallographyREVIEWER COMMENTS

Reviewer #1 (Remarks to the Author):

In their manuscript, Maity et al investigate the Carbon monoxide (CO) release upon photoexcitation of a Mn(CO)₃ unit in the reaction center of a HEWL protein by means of serial femtosecond crystallography. Scattering patterns are extracted from experiments at a free-electron laser to reconstruct the electron density at the reaction center and follow the spatial geometry changes of the Mn(CO)₃ unit at 10ns, 100ns and 1μs after photexcitation with a 365nm laser pulse. The results are combined with QM/MM simulations to rationalize the observed changes in electron density.

The main novelty of this work stems from the incorporation of a small synthetic molecule into a protein matrix to observe its light-induced dynamics, in contrast to conventional SFX-studies on reactions in naturally occurring biomolecules. While the idea is promising, I find the manuscript not suitable for publication yet due to the reasons given below. I encourage the authors to re-submit the manuscript after major revision and once additional experiments and simulations are performed, should the findings hold.

1) The temporal resolution of the experiments must be significantly increased. The main claim of the study, as stated in lines 33 and 34, is that "we succeeded in capturing the real-time formation of Mn-carbonyl intermediates during the CO release reaction" In the first scattering pattern at 10ns in Fig. 4, the axial CO is already dissociated, and no intermediate is observed.

2) The term serial femtosecond crystallography is misleading. While the X-ray probe pulse has a low-femtosecond temporal width, the temporal resolution of this study is in the high nanosecond regime, prohibiting "real-time investigation of reaction intermediates", as claimed by the authors. A many-ns-temporal resolution also does not qualify for "ultra-short intervals" especially with respect to fs crystallography. I also find the term time-resolved-SFX misleading, since serial femtosecond crystallography should be inherently time-resolved.

3) Considering a recent method review on photo-induced SFX by one of the leading researchers in the field (<https://doi.org/10.1038/s43586-022-00141-7>), I find that several best-practices-criteria are not met. In particular (partly quoting from the review article):

(i) It should be ensured that only the intended reaction should be initiated. Populating non-productive pathways possibly results in artefacts that may be mistaken for physiological structural changes.

(ii) It is highly desirable to establish online monitoring to determine what SFX is probing. A good example is the simultaneous SFX data collection and monitoring of metal oxidation states by X-ray emission spectroscopy. This is especially relevant since in tr-spectroscopy literature (e.g. doi.org/10.1021/jz302061q and [10.1016/j.ica.2011.10.047](https://doi.org/10.1016/j.ica.2011.10.047)), it is discussed that the dissociation of CO groups beyond the first one is usually accompanied by an oxidation of the metal center.

(iii) A power titration should be performed where the magnitude of the photoproduct signal is plotted as a function of laser energy. Deviation from a straight line through the origin indicates that too much energy was absorbed by the system, populating non-productive pathways, possibly resulting in artefacts that may be mistaken for physiological structural changes.

(iv) Knowledge of reaction kinetics in crystal is essential, e.g. to determine the ideal time stamps at which SFX data should be collected.

It is essential that the authors perform supporting experiments targeted at the above points to corroborate their findings and solidify their discussion.

4) There is a considerably body of time-resolved spectroscopy literature investigating the CO-release of metal carbonyls in gas phase and solution. One recent well-executed example on a Mn(CO)₃-unit is doi.org/10.1021/jz302061q. These already find the formation of monocarbonyl on the low picosecond timescale. A more extensive discussion of the author's data and findings to the time-scale, intermediates and reaction products observed in such experiments would be helpful to highlight the information gain by the author's study.

5) More rigorous connection of QM/MM data to the difference maps is needed. They are discussed mostly separately. For example: Fig.4di indicates that bond-distances between the metal center and COeq2 are significantly changed. Do the QM/MM optimizations of structures support this?

6) There is a quite interesting dependence on the intensity of the pump laser pulse in Fig. 4c iv versus Fig. 4c v, where in the latter a second CO seems to dissociated, whereas in the former it

does not. This further highlights the necessity of a power titration to ensure both experiments stay within the linear regime. It further invites a discussion in comparison with the article doi.org/10.1021/jz302061q, finding that "re-excitation with a second UV pulse did not lead to observable monocarbonyl species".

7) A very pronounced, negative density at Mn in Fig. 4 is only hand-waivingly discussed as "Potential release of whole MnCO₂ unit to some extent", while in the same figure, same or less density difference at CO is discussed as "definite proof of CO release". This needs more distinguishment/clarification.

8) Fig. 4b is confusing and, to my eye, not fully clarifying the pulse configuration in the experimental setup.

9) Some quantities and terms in the manuscript are not properly explained or rigorously defined. These are for example:

- FO-FC-map

- σ

- B-Factor

- Nudged elastic band method

Please make sure to properly define (formula), reference and/or explain each term, parameter and quantity.

10) In the QM/MM simulations, different basis sets are used for different calculations. Please explain why, potentially with benchmarking a smaller/lower basis set wherever deviation is made.

11) What is the electronic character of the excited states involved in the photochemistry? This should be readily accessible from the simulations and could rationalize the CO-dissociation.

12) How are the excited state barriers exactly calculated?

13) It is stated in the manuscript that the entire QM system is shown in Fig. S11. Compared to this, Fig. 5 has additional water in the QM region, whereas the dissociated CO is not shown. Where does it go, and how is the exchange of groups in and out of the QM region treated across all calculations?

Reviewer #2 (Remarks to the Author):

The manuscript "Real-time observation of a metal complex-driven reaction intermediate using a porous protein crystal and serial femtosecond crystallography" is demonstrating that TR-SFX allows real-space imaging of the release of CO from a Mn(CO)₃ reaction center in the nano second time regime in a protein environment. The authors identified intermediates of the light-induced reaction and structural details, which are hardly accessible by other experimental techniques. The experimental data is complemented and verified using quantum mechanical calculations.

Background and principles are well explained and the experimental design with required control experiments is fine. The experimental idea is interesting and detection of the reaction intermediates in conjunction with QM/MM calculations is sound. The authors also indicate influences of the protein environment on the reaction, which is quite important.

My major concern is related to the applicability of the selected model system to other small molecule reactions under similar conditions. The reasons for selecting lysozyme here likely include that it has been shown before to bind an engineered Mn(CO)₃ metal center and lysozyme crystal suspensions can rather easily be prepared. However, I would see a broader applicability of the method when trying to go to a metal-organic framework (MOF), a "crystalline sponge" (e.g. <https://doi.org/10.1107/S2052252515024379>) or a crystalline protein scaffold with even higher solvent content than the selected lysozyme as reaction environment. This might also allow to minimize the influence or bias by the individual asymmetric protein environment. The results obtained for the CO release are sound and valuable, but if I understood correctly that the experiment and the reaction studied is considered a model system, I am not yet convinced about a broad applicability. I suggest to discuss or verify which other reactions could be studied in a similar protein environment and which potential one or another (protein) environment presumably has. This seems to be closely connected to the potential impact of this approach in the near-future in comparison to previous TR-SFX experiments.

Are there alternative ways (other than light) at the moment to efficiently trigger a similar reaction in a similar TR-SFX approach?

Some statements ("...we have established that the methodology is useful for exploring difficult reaction mechanisms..."; "The method also offers the useful option to control a given reaction by modifying the reaction environments by changing suitable amino acid residues..."; "Our demonstration might be useful for studying structural dynamics under external triggers, new bond formation or bond-breaking processes.") are too vague or too general in my opinion and the discussion part should be rephrased and extended accordingly. Is it possible to be more specific which amino acid exchanges on the surface of lysozyme would be useful based on the structures? Some parts of the rather short discussion section seem to overlap a bit with previous sections of the paper and I would like to encourage optimization of the whole section in a revised version.

Minor additional points of concern:

- Figure 1: Both crystal lattice images are showing similar things. Is there a way to combine them?
- Figure 2a: The two crystal micrographs might be hard to compare because of different background and/or optics settings? I would anticipate a clearer change during the immobilization as indicated by the two reaction tube pictures.
- Tables S1A/S1B/S1C: I did not see the PDB reports for the five structures uploaded to the PDB, but table S1 should be checked for mistakes: I think that the unit cell c axis of 8WZT is not 81.7? And "a = b = c" should be changed to "a, b, c", the same for the angles. And the R-work/R-free format of 8WZT should be corrected.

Minor formal issues:

- Line 91: "...during the CO release reaction could be characterized [word missing?] TR-SFX."
- Line 135: MLCT should be spelled in full as metal-to-ligand charge-transfer
- Line 397: "it is believed that this [word missing?] possible because of the asymmetric protein environment."
- Supporting Information, line 27: were purchased
- Supporting Information, line 113: "...and deposited [words missing?] wwPDB server..."

Reviewer #3 (Remarks to the Author):

In the manuscript titled "Real-time observation of a metal complex-driven reaction intermediate using a porous protein crystal and serial femtosecond crystallography" by Maity et al. the authors present a study on a CO-releasing manganese complex by incorporation into lysozyme crystals and probing the time-resolved reaction by serial femtosecond crystallography at SACLA.

Presented is an interesting approach for facilitating the TR-study of a small molecule by incorporation into a large unit cell protein crystal, yielding indexable diffraction patterns with serial "snapshot" data collection at an FEL. The results from the electron density interpreted reaction intermediates are then further analyzed with QM/MM simulations. The technique itself is innovative and interesting and, with improved experimental rigor, would be worthy of publication in Nature Communications.

Some details regarding the experimental conditions are lacking, which could potentially impact the interpretation of results and conclusions drawn. Furthermore, the authors elude to inconsistencies in time points, again without sufficient experimental detail to evaluate their reasoning for this. Detailed concerns and suggestions are outlined below, along with minor comments.

Major concerns:

More detailed analysis on whether the interleaved dark structures for time points >17ms is required based on the experimental conditions. What were the flowrates, jet diameter etc.? In a recent study by Li et al. (<https://doi.org/10.1107/S2052252521002177>), the authors found that they severely underestimated the required sample flowrate to adequately avoid light contamination in a similar high-viscosity jet set-up at SACLA.

If assuming similar condition for the experiment presented here to that published by Li et al. (since these are missing from this manuscript); i.e. Jet diameter: 125µm, while collecting at 30Hz:

For the stated laser spot size of 100 μ m FWHM, an absolute theoretical minimum flowrate of 3 μ L/min would have been required to avoid light contamination. However, when considering the observations by Li et.al. that the optical illumination area is scattered well beyond the area defined by the laser spot size, they required a flowrate of 9.8 μ L/min to sufficiently replenish sample in between pulses with a 250 μ m laser spot size (top-hat, not FWHM!). So I would expect similar requirements for the data presented here.

How were the longer time delays achieved? By offsetting pump and probe in space? What were the parameters to achieve this? Was jet speed measured or estimated? The 1ms delay difference maps presented seems to be light contaminated (Figure S9b). The 17ms does not (Figure S9c). Was the crystal suspension not washed prior to the SFX experiment, as for the UV-vis and ATR-IR spectral analysis? If not, what was the rationale if it was deemed necessary for the spectral analysis? Without sufficiently describing the experimental conditions this reviewer finds it hard to accept the authors' speculation that the inconsistencies in electron density result from replenishing compound from the bulk solvent. Light contamination seems more likely.

How were the 2 different laser fluences chosen? Is it conceivable that 40 μ J is heating the sample, resulting in the overall blurring of the 2Fo-Fc maps (Figure 4c) and higher B-factors for the protein and solvent (Table S1B)? This reviewer would have liked to see a more detailed correlation analysis of the QM/MM results (Energy barriers presented in Figure 5b) and the pumping scheme/laser fluence employed. How many photons per reaction center?

Why is there such a noticeable discrepancy in the unit cell dimensions across the datasets (TableS1A, 80.8, 80.8, 37.6 for 1 μ s 40 μ J; 81.0, 81.0, 37.7 for Complete Darkness and 10 ns 20 μ J; 81.7, 81.7, 37.9 for 100ns 20 μ J and 1 μ s 20 μ J)? Isomorphous difference maps are very sensitive to even minute differences in unit cell dimensions. Since the interleaved "dark" datasets match their respective "light" counterparts (and assuming they are not light contaminated) this is unlikely caused by the reaction; is it conceivable that the injector was not always installed in the same position (in which case the detector to sample difference should probably be tweaked instead of the differing unit cell constants)?

Some of the conclusions are over-generalized blanket statements, and the reader would benefit from more concrete specifications for this to be a widely applicable new method:

- 22 \AA is not very large for a protein solvent channel. What implications does this have on systems that can be studied using HEWL.
- HEWL was crystallized in its tetragonal isoform, despite the orthorhombic form having larger solvent channels. The authors rationalize this with the tetragonal form being easier to crystallize homogeneously with high yield. This is a reasonable argument when HEWL crystals are regarded as a readily accessible scaffold material. However, then arguing that the ligand environment can be easily modified by mutating the surrounding amino acids completely contradicts this. Cloning, recombinant expression, purification and then crystallizing a protein (even lysozyme) is a tedious and expensive endeavor. This reviewer would be more interested in the generalizability of this approach. E.g. what ligands would be amendable to studying using this technique. Or is it the uniqueness of this particular reaction ("After CO release, the Mn center will typically undergo dimerization, aerial oxidation or precipitation in solution and such results complicate mechanistic investigations") that make it amendable to this technique? Are there a larger group of compounds that would bind to HEWL and lend itself to TR-SFX study?

Minor comments:

I'm not sure what the value of Figure 1a is. As ubiquitous as tetragonal HEWL crystals are, and protein crystals having solvent channels being common knowledge, this figure, at best, belongs in the supplement. Figure 3 has the same information, plus the more interesting depiction of the precise binding site of Mn(CO)₃ to the HEWL molecules, and the associated accessibility via the solvent channel.

Figure 2: Comparison spectra for apo-HEWL and post-photolysis would make for a nice addition. Also, c) has "au" as y-axis units whereas figure b) doesn't have units. The distinction may be arbitrary, but consistency would be nice.

Figure 3c) replace the 20 "uJ" with "μJ"

Figure 5b) what are the units of delta E?

Reference 33 is an odd unconventional choice. I suggest citing the original work by Chapman et al (<https://doi.org/10.1038/nature09750>) and Neutze et al. (<https://doi.org/10.1038/35021099>).

Supplement: XFEL Data processing: What is meant by "merging strategies were optimized"? What parameters and sub-programs were ultimately used for the data presented? Were Friedel's merged etc.?

Figure S7: Why would the manganese occupancy differ from the COs, unlike for Figure S8 where they are consistent with the COs and the H2Os? Either the MnCO3 complex is bound in all binding sights or not 10ns after illumination. This does not match the discussion that the COs are consecutively replaced by H2Os.

Language: While readable and understandable, the manuscript would benefit from a general language improvement with more concise terminology, e.g.:

P 3 | 59: "over a wide time range from femtoseconds" -> To? From femtoseconds is not a range.

P7 | 120 & P 9 | 152: "A single diffraction was collected per crystal" -> A single diffraction pattern was collected per crystal. Plus, this is not substantiated without stating flow rates!

P9 | 153: "and a number of diffraction was collected" -> a minimum of XYZ indexable diffraction patterns were collected per dataset

P17: l. 208: "this suggests that the axial CO ligand is more sensitive to light" -> A CO ligand is not light-sensitive to my knowledge.

Response to Reviewers

Reviewer #1:

In their manuscript, Maity et al investigate the Carbon monoxide (CO) release upon photoexcitation of a Mn(CO)₃ unit in the reaction center of a HEWL protein by means of serial femtosecond crystallography. Scattering patterns are extracted from experiments at a free-electron laser to reconstruct the electron density at the reaction center and follow the spatial geometry changes of the Mn(CO)₃ unit at 10ns, 100ns and 1 μ s after photoexcitation with a 365nm laser pulse. The results are combined with QM/MM simulations to rationalize the observed changes in electron density.

The main novelty of this work stems from the incorporation of a small synthetic molecule into a protein matrix to observe its light-induced dynamics, in contrast to conventional SFX-studies on reactions in naturally occurring biomolecules. While the idea is promising, I find the manuscript not suitable for publication yet due to the reasons given below. I encourage the authors to re-submit the manuscript after major revision and once additional experiments and simulations are performed, should the findings hold.

R1-1: 1) The temporal resolution of the experiments must be significantly increased. The main claim of the study, as stated in lines 33 and 34, is that “we succeeded in capturing the real-time formation of Mn-carbonyl intermediates during the CO release reaction” In the first scattering pattern at 10ns in Fig. 4, the axial CO is already dissociated, and no intermediate is observed.

Reply1-1: We agree with the reviewer that higher temporal resolution is better for exploring the detailed reaction mechanism. In a CO release reaction, exchange of CO ligands with water molecules after photoexcitation yields an aquated metal complex which is the reaction intermediate. The fact that the CO is about to leave is neither a stable intermediate nor a transition state, so, we do not think it can be captured by TR-SFX in principle. Therefore, determining the real-space structure of the transition state involving the departing CO ligand and incoming water molecule together (for seven coordinated associated mechanisms) is challenging. So far such short-

lived transition states are achieved only for conformational changes in organic molecules in natural proteins. Such transition states in natural metalloenzyme reactions were also not observed before. As mentioned by the reviewer, we were also interested in capturing those transition states but found it difficult to observe by ns laser.

We included such an explanation in the revised manuscript (Page 12-13, Line 224-227).

R1-2: 2) The term serial femtosecond crystallography is misleading. While the X-ray probe pulse has a low-femtosecond temporal width, the temporal resolution of this study is in the high nanosecond regime, prohibiting “real-time investigation of reaction intermediates”, as claimed by the authors. A many-ns-temporal resolution also does not qualify for “ultra-short intervals” especially with respect to fs crystallography. I also find the term time-resolved-SFX misleading, since serial femtosecond crystallography should be inherently time-resolved.

Reply1-2: “Serial femtosecond crystallography” is a standard term in the field. “Femtosecond” relates to the width of the probing X-ray pulse, not the temporal resolution of the time-resolved study.

“serial femtosecond crystallography should be inherently time-resolved”: No. Many SFX studies are carried out without any reactions to study a static structure without radiation damage (<https://doi.org/10.1038/s41586-021-04218-3>; <https://doi.org/10.1073/pnas.1517770113>)

R1-3: 3) Considering a recent method review on photo-induced SFX by one of the leading researchers in the field (<https://doi.org/10.1038/s43586-022-00141-7>), I find that several best-practices-criteria are not met. In particular (partly quoting from the review article):

(i) It should be ensured that only the intended reaction should be initiated. Populating non-productive pathways possibly results in artefacts that may be mistaken for physiological structural changes.

Reply1-3: The HEWL-Mn(CO)₃ contains only one reaction center, and the corresponding light-induced CO release reaction was intended to be studied in this manuscript. The CO release reaction by Mn-carbonyl is a well-known reaction that was previously studied both spectroscopically and computationally. Real-space structure determination of photoproducts by TR-SFX was performed in this study. Regarding the non-productive pathways, we assigned the intermediates based on difference density features consistent with the QM/MM calculations. The possibility of formation

of other intermediates in small fractions cannot be completely excluded as they have little high energy barrier compared to the assigned stable intermediates (as shown in a flow chart of Figure S17). However, assigning the low-populated intermediates from electron density maps is challenging. Thus, the reaction path shown in the manuscript is the major one. Such an explanation was included in the revised manuscript (Page 23, Line 428-432).

R1-4: (ii) It is highly desirable to establish online monitoring to determine what SFX is probing. A good example is the simultaneous SFX data collection and monitoring of metal oxidation states by X-ray emission spectroscopy. This is especially relevant since in tr-spectroscopy literature (e.g. doi.org/10.1021/jz302061q and 10.1016/j.ica.2011.10.047), it is discussed that the dissociation of CO groups beyond the first one is usually accompanied by an oxidation of the metal center.

Reply1-4: We agree with the reviewer that simultaneous monitoring of SFX data collection and changes in metal oxidation state by spectroscopy would be useful in exploring the detailed reaction mechanism. However, it is challenging to combine both when studying the reaction in streaming crystals embedded with high-viscosity carrier media. Although Yano, J. and Yachandra, V. K. et al. established on-line spectroscopy with their tape drive setup (<https://doi.org/10.1016/j.sbi.2023.102604>), simultaneous on-line spectroscopy has not been successful in the system with high-viscosity sample injectors. In addition, the CO release reaction by the $\text{Mn}(\text{CO})_3$ complex is a well-known reaction and has been studied previously in solution by transient spectroscopy (doi.org/10.1021/jz302061q and 10.1016/j.ica.2011.10.047) as pointed out by the reviewer. Therefore, changes in the metal oxidation state and water exchange during the reaction are known. As pointed out by the reviewer, the oxidation of metal center after CO dissociation is included in the revised manuscript (Page 15, line 299-300).

R1-5: (iii) A power titration should be performed where the magnitude of the photoproduct signal is plotted as a function of laser energy. Deviation from a straight line through the origin indicates that too much energy was absorbed by the system, populating non-productive pathways, possibly resulting in artefacts that may be mistaken for physiological structural changes.

Reply1-5: As requested by the reviewer, we performed a power titration at 10 ns and 1 μ s delay, at which the laser intensity was varied and plotted against the negative difference density features (page S4, line 124-128; Figure S9A, S9B, and Table S4). It was found that the difference density features increased gradually with laser power. In addition, we plotted the apparent occupancy (based on omit map) of the released CO or Mn with laser intensity, which showed a linear increase up to 20 μ J and deviated from linearity. This indicates the influence of higher laser intensity. We explained this in the revised manuscript (Page 14-15, Line 269-279). The quantification method has been given in SI (page S4, line 124-128).

R1-6: (iv) Knowledge of reaction kinetics in crystal is essential, e.g. to determine the ideal time stamps at which SFX data should be collected.

It is essential that the authors perform supporting experiments targeted at the above points to corroborate their findings and solidify their discussion.

Reply1-6: We agree with the reviewer that when exploring a reaction mechanism, a study of reaction kinetics is essential to understand more. However, studying reaction kinetics in a crystalline state is challenging. In spectroscopy with crystals, a single crystal is mounted to a quartz cell along with carrier media and measured, which requires a special setup for the measurement. If signals from a crystal are repeatedly accumulated, such as the photocycle of bacteriorhodopsin (*Science*, 2016, 354(6319):1552-1557), the experiment is feasible, but if it is an irreversible reaction, it is quite challenging since a weak signal from a single crystal is needed to be collected by exchanging it for a new crystal. For example, Ilme Schlichting, Thomas Barends, and coworkers have reported TR-SFX studies of carboxy-myoglobin (Barends et al. *Science*, 2015. DOI: 10.1126/science.aac5492). However, their reports do not validate the reaction course in crystals by other methods, such as spectroscopy, since the target reaction is irreversible. In general, it is very difficult to trace reactions such as CO dissociation in crystals. Therefore, in such cases, it is often combined with computational approaches, as we also conducted in this study.

R1-7: 4) There is a considerable body of time-resolved spectroscopy literature investigating the CO-release of metal carbonyls in gas phase and solution. One recent well-executed example on a Mn(CO)₃-unit is doi.org/10.1021/jz302061q. These already find the formation of monocarbonyl on the low picosecond timescale. A more extensive discussion of the author's data and findings to

the timescale, intermediates and reaction products observed in such experiments would be helpful to highlight the information gain by the author's study.

Reply1-7: We agree with the reviewer that there is a considerable body of time-resolved spectroscopy literature investigating the CO-release of metal-carbonyls in gas phase in solution. A comprehensive summary of these studies reveals that the initial release of CO and replacement of the vacant coordination site by solvent is a common occurrence and detected by ultra-fast spectroscopy (doi.org/10.1021/jz302061q; doi.org/10.1021/acs.inorgchem.9b02758). Subsequent releases of other CO ligands are linked to the oxidation of the Mn(I) center, resulting in dimerization, possibly due to the release of strongly π -accepting CO ligands (doi.org/10.1016/j.ica.2011.10.047; doi.org/10.1039/C6DT02020H). The final Mn-dimer structure is typically determined through X-ray crystallography. However, the determination of structures following the 1st, 2nd, or 3rd CO releases relies on spectroscopic evidence. In contrast, our study directly determines the structures after the 1st and 2nd CO releases using the TR-SFX method.

As the reviewer pointed out, we added more discussion on this in the revised manuscript (Page 13, line 229-232, Page 15, Line 299-300, Page 23, Line 415-432).

R1-8: 5) More rigorous connection of QM/MM data to the difference maps is needed. They are discussed mostly separately. For example: Fig.4di indicates that bond-distances between the metal center and COeq2 are significantly changed. Do the QM/MM optimizations of structures support this?

Reply1-8: It is to be noted that Figure 4d-i did not represent the bond-distance between Mn-COeq2 for 10ns/20 μ J data. As mentioned in the Figure 4 caption, it shows the difference density features for 10ns/20 μ J and the given complete darkness model (stick model) shown in the figure was used for phase when calculating the difference map. The nature of the difference map indicated the movement of the Mn atom and lies close to the red (negative) map. When overlapping (Figure S11d) both structures (Complete Darkness vs. 10ns/20 μ J), we observed a movement of 0.098Å. This is now mentioned in the revised manuscript (Page 13, line 234-235).

We added the QM/MM data with difference density features obtained by TR-SFX in the revised manuscript (Table S5 and S6). We are now correlating with the QM/MM calculation. Structural

parameters are summarized in Table S5. We also compared based on the root-mean-square deviation (RMSD) for the Mn center in Table S6. These results indicate that QM/MM optimized structures (1, 2, 3) correspond to each TR-SFX structure (8WZF, 8WZG, 8WZV). We added the following discussion in the revised manuscript on page 19-20, line 369-378.

“This QM/MM expected reaction (1→2→3→4) is consistent with the reaction observed in the TR-SFX measurements. From a structural point of view, elongations of the Mn-ligand bond length upon the CO release are also consistent (Table S5), and the Mn center of the structure in complete darkness, 10ns and 1μs delay becomes closest to one of **1**, **2** and **3** of QM/MM optimized structures based on the values of root-mean-square deviation (RMSD) (Table S6).”

R1-9: 6) There is a quite interesting dependence on the intensity of the pump laser pulse in Fig. 4c iv versus Fig. 4c v, where in the latter a second CO seems to dissociated, whereas in the former it does not. This further highlights the necessity of a power titration to ensure both experiments stay within the linear regime. It further invites a discussion in comparison with the article doi.org/10.1021/jz302061q, finding that “re-excitation with a second UV pulse did not lead to observable monocarbonyl species”.

Reply1-9: As pointed out by the reviewer, we performed a power titration experiment, as shown in Figure S9A and S9B in the revised SI and explained in the reply of R1-5. The 40 μJ data revealed the release of 2nd CO at this stage. However, the density feature did not follow the linear relationship. At 40 μJ, the density feature increased, indicating potential release at this stage, whereas the same for CO_{ax} increased slightly.

We appreciate the referee’s idea of re-excitation with 2nd UV pulse. However, since the power titration at higher laser intensity is not linear and also the difference density features at longer delay time (1 ms and 17 ms in Figure S10) did not show ascending order, we think re-excitation might not give any new information except stepwise CO release instead it might lead to multiple intermediate and complicated correct assignments of intermediates. Therefore, we can conclude that our method is valid for the initial stages of the reaction involving two CO releases and that the study becomes complicated at later stages of the reaction (Page 15, Line 291-292).

R1-10: 7) A very pronounced, negative density at Mn in Fig. 4 is only hand-waivingly discussed as “Potential release of whole MnCO₂ unit to some extent”, while in the same figure, same or less density difference at CO is discussed as “definite proof of CO release”. This needs more distinguishment/clarification.

Reply1-10: As the reviewer raised the concern over the prominent negative density feature at Mn, we included an explanation in the revised manuscript (Page 13, Line 238-241, 243-245).

The decrease in Mn occupancy might be caused by releasing the whole Mn-carbonyl fragment. As shown in Figure S17, although the reaction is expected to follow the lowest energy barrier route involving intermediates, the energy barrier to release the whole Mn-carbonyl fragment (Imidazole release) from the protein is higher but not very large. Therefore, a small fraction of the whole Mn-carbonyl might be released at the same time during the formation of the intermediate.

It is known that the Mn ion gets oxidized after the first CO release, which causes changes in the electronic environment. This might give the negative density feature in the difference map. (<http://dx.doi.org/10.1039/C6DT02020H>)

R1-11: 8) Fig. 4b is confusing and, to my eye, not fully clarifying the pulse configuration in the experimental setup.

Reply1-11: Figure 4b is the standard experimental setup followed in most of the TR-SFX reports. This setup explains the setting of the XFEL pulse and pump laser pulse and how the delay time was calculated. The 30Hz repetition was used for the XFEL pulse, and 10Hz repetition was used for the pump probe pulse (except the 100 ns data). To avoid confusion, the figure caption has been modified in the revised manuscript (Page 17, Line 313-315).

R1-12: 9) Some quantities and terms in the manuscript and not properly explained or rigorously defined. These are for example:

- FO-FC-map
- σ
- B-Factor
- Nudged elastic band method

Please make sure to properly define (formula), reference and/or explain each term, parameter and quantity.

Reply1-12: Fo-Fc map, sigma level, and B factor are standard textbook terms in crystallography. We further mention in the experimental section (Page S4, line 109-112, and 125). We cited references (42 and 43) in the main text for the Nudged elastic band method which is a technique used in QM/MM calculations to study reactions, particularly those occurring on complex surface or in condensed phases.

R1-13: 10) In the QM/MM simulations, different basis sets are used for different calculations. Please explain why, potentially with benchmarking a smaller/lower basis set wherever deviation is made.

Reply1-13: The validity of the computational conditions such as basis set, DFT functional, and QM region for the structure in the ground state and the energy of the first excited state are summarized in the Supporting Information (Table S7). The RMSD values of the Mn center by using larger basis sets are 0.228Å, 0.216Å, and 0.216 Å compared to the X-ray structure (Figure S13). This result means that the geometrical structures are only improved by 0.008 Å, even for the larger basis set. On the other hand, the excited state property, such as peak-top wave number around 400 nm, is very sensitive to the basis sets (384(entry 2) – 387(entry 1) = - 3 nm), DFT functional (389 (entry 6) – 386 (entry 3) = +3 nm) and QM region (394(entry 9)- 389(enyty6) = 5 nm). Even though the optimized geometry by using TZVP(B2) does not improve for the peak-top wavelength compared to the structures calculated by using DZVP (394(entry12)-394(entry9) = 0 nm), considering the computational costs, large number of structural optimization should be performed at the B3LYP/DZVP(B1) theoretical level, while the UV spectra calculations could be carried out at the larger basis set and larger QM region as the same level of entry 9. We included these in the revised main text (page 18, line 337-341).

We added the above discussion in the Supporting Information (Page S6 in SI, line 178-192).

R1-14: 11) What is the electronic character of the excited states involved in the photochemistry? This should be readily accessible from the simulations and could rationalize the CO-dissociation.

Reply1-14: Molecular orbitals mainly contributing to the photoexcitation to the first excited states are shown in Figure S16 (page S22). The HOMOs are mainly composed of Mn d orbitals, while the LUMOs are more delocalized over the ligands with antibonding character. This feature is

consistent with the photochemical dissociation of ligands in their excited states. However, orbital distributions over ligands in LUMOs are not specific only to the easiest dissociation ligand. Therefore, the molecular orbital analysis is not fully informative for the specific photochemical reactions of the HEWL-Mn(CO)₃Wat₂.

We added a following discussion in the revised manuscript in page 19, line 364-368.

“Molecular orbitals contribute to the first excited state as shown in Figure S16. The highest occupied molecular orbitals (HOMOs) are mainly distributed on the Mn d orbitals, and the lowest unoccupied molecular orbitals (LUMOs) are more delocalized to the CO and water ligands with antibonding character. These MO transitions also support the enhancement of the ligand dissociation via the excited states.”

R1-15: 12) How are the excited state barriers exactly calculated?

Reply1-15: We searched by following the ligand water exchange pathway in the ground state using the nudged elastic band (NEB) method. Potentials in excited states are drawn by calculating the excited energies above the ground state. These virtual excitation potentials tend to overestimate the reaction barriers because the excited state barriers are fully optimized. Full path optimization in excited states is not implemented in the available NWChem program package. Even though the approximation was introduced, energy barriers in the excited states are significantly decreased compared to the ground state by 14.0 kcal mol⁻¹, 13.9 kcal mol⁻¹, and 11.6 kcal mol⁻¹ for the first CO release, the second CO release, and Mn center release, respectively. We added the energy profiles of the important reaction steps in the supporting information to see the barriers in the excited states (page 19, line 356-358; Figure S15 and S17).

R1-16: 13) It is stated in the manuscript that the entire QM system is shown in Fig. S11. Compared to this, Fig. 5 has additional water in the QM region, Whereas the dissociated CO is not shown. Where does it go, and how is the exchange of groups in and out of the QM region treated across all calculations?

Reply1-16: Depending on the reactions, QM regions which should be treated in the quantum mechanical method are changed. Dissociated CO ligands to solvent are excluded from the QM/MM system in latter reactions. In this revision, we updated results by using a larger QM region

of the QM/MM calculations. Explicit QM regions which used for UV spectra calculations are shown in Figure S14 and mentioned in the revised text and SI (page 22, line 410-411; page S5, line 161-162).

We added a following sentence in the supporting information (Page S5, line 154-155).

“The nearest solvent water molecule was included in the QM region depending on the reaction steps (Figure 5).”

Reviewer #2:

The manuscript “Real-time observation of a metal complex-driven reaction intermediate using a porous protein crystal and serial femtosecond crystallography” is demonstrating that TR-SFX allows real-space imaging of the release of CO from a Mn(CO)₃ reaction center in the nano second time regime in a protein environment. The authors identified intermediates of the light-induced reaction and structural details, which are hardly accessible by other experimental techniques. The experimental data is complemented and verified using quantum mechanical calculations.

Background and principles are well explained and the experimental design with required control experiments is fine. The experimental idea is interesting and detection of the reaction intermediates in conjunction with QM/MM calculations is sound. The authors also indicate influences of the protein environment on the reaction, which is quite important.

R2-1: My major concern is related to the applicability of the selected model system to other small molecule reactions under similar conditions. The reasons for selecting lysozyme here likely include that it has been shown before to bind an engineered Mn(CO)₃ metal center and lysozyme crystal suspensions can rather easily be prepared. However, I would see a broader applicability of the method when trying to go to a metal-organic framework (MOF), a “crystalline sponge” (e.g. <https://doi.org/10.1107/S2052252515024379>) or a crystalline protein scaffold with even higher solvent content than the selected lysozyme as reaction environment. This might also allow to minimize the influence or bias by the individual asymmetric protein environment. The results obtained for the CO release are sound and valuable, but if I understood correctly that the experiment and the reaction studied is considered a model system, I am not yet convinced about a broad applicability. I suggest to discuss or verify which other reactions could be studied in a similar protein environment and which potential one or another (protein) environment presumably has. This seems to be closely connected to the potential impact of this approach in the near-future in comparison to previous TR-SFX experiments.

Reply2-1: We thank the reviewer’s thoughtful consideration and insightful feedback on the applicability of our system and potential alternatives. As you pointed out, the introduction and discussion of the manuscript already touch upon the uses and applicability of the studies described.

In response to your valuable input, we further elaborated on these aspects in the revised manuscript to provide additional clarity (Page 24, Line 438-440, and 449-456).

We appreciate your interest in exploring the broader applicability of our methodology, especially in the context of metal-organic frameworks (MOF), crystalline sponges, and other protein scaffolds with larger pore sizes and higher solvent content. While other large-pore crystalline protein scaffolds are suitable, there is no established method to use MOF crystals in SFX due to difficulty in structure determination. This is attributed to the complexity of determining the structure of metal complex crystals by SFX, which is characterized by fewer diffraction spots than protein crystals with large unit cells, making indexing challenging. Additionally, metal complex crystals are tightly packed with low solvent content, potentially hindering reactions within the crystal. Moreover, structural changes during reactions might alter lattice, making it challenging to directly compare the structure before and after the reaction. Such explanations are included in the revised manuscript (Page 24, Line 438-440), and the difficulties are described in the introduction section (Page 4, Line 74-78).

Regarding the “influence or bias by protein environment,” each protein environment is unique and affects reactions differently. The protein environment surrounding the reaction center in HEWL is unique and will be different if other protein scaffolds are used. Thus, the nature of the reaction like the order of CO release, etc., might be changed due to changes in the environment. It is possible to change the protein environment (not for HEWL, as it is hard to express and purify) to control the intermediate trapping. This is also included in the revised manuscript (Page 24, Line 449-456). As requested by the reviewer, we have included a discussion on potential reactions and possibilities that could be explored for broader applicability (Page 25, Line 467-472). Furthermore, we agree with your observation that our work is closely connected to the potential impact of this approach in the near future compared to the previous TR-SFX experiments.

R2-2: Are there alternative ways (other than light) at the moment to efficiently trigger a similar reaction in a similar TR-SFX approach?

Reply 2-2: Yes, there are alternate ways to trigger a reaction other than light. However, in those studies, the time resolution is poor and needs to be improved to apply the method to ns-order elementary reactions such as the one we used in this manuscript. For example, a mixed jet system could be used in which crystals are mixed with another solution (diffusive mixing) by TR-SFX in

several millisecond intervals. The methodology is applied for studying substrate binding into enzymes in a time-resolved manner as well as exploring chemical reactions (<https://doi.org/10.1038/s41467-021-24757-7>, <https://doi.org/10.1021/jacs.3c04991>). In addition, the temperature of the crystals can be raised gradually to observe time-resolved changes in the protein and might be applicable to study changes in a reaction center (<https://doi.org/10.1038/s41557-023-01329-4>).

This is included in the revised manuscript (Page 25, Line 467-472).

R2-3: Some statements (“...we have established that the methodology is useful for exploring difficult reaction mechanisms...”; “The method also offers the useful option to control a given reaction by modifying the reaction environments by changing suitable amino acid residues...”; “Our demonstration might be useful for studying structural dynamics under external triggers, new bond formation or bond-breaking processes.”) are too vague or too general in my opinion and the discussion part should be rephrased and extended accordingly. Is it possible to be more specific which amino acid exchanges on the surface of lysozyme would be useful based on the structures?

Reply2-3: According to the reviewer’s comment, to eliminate any potential confusion, we have rephrased and extended the discussion by incorporating specific examples (Page 5, line 109-110; page 25, line 483-484 and page24, line 449-456).

Regarding the amino acid exchange, since lysozyme expression in E. coli is difficult, it is applicable when other protein scaffolds, which were added as references in the revised text (Page 24, Line 449-456), will be used. Therefore, we rephrased the text from “The method also offers the useful option to control a given reaction by modifying the reaction environments by changing suitable amino acid residues...” to “The method hold promises to explore reaction mechanism of artificial metalloproteins and metalloenzymes.” (Page 25, Line 483-484).

R2-4: Some parts of the rather short discussion section seem to overlap a bit with previous sections of the paper and I would like to encourage optimization of the whole section in a revised version.

Reply2-4: We rechecked the manuscript to avoid any repetition (Page 23, Line 428-432).

Minor additional points of concern:

R2-5A:- Figure 1: Both crystal lattice images are showing similar things. Is there a way to combine them?

Reply 2-5A: Although they are same as in Figure 1a, one is the top view, and the other is the side view. We have modified Figure 1a.

R2-5B:- Figure 2a: The two crystal micrographs might be hard to compare because of different background and/or optics settings? I would anticipate a clearer change during the immobilization as indicated by the two reaction tube pictures.

Reply 2-5B: We replaced the pictures having same background for both in Figure 2a.

R2-5C: Tables S1A/S1B/S1C: I did not see the PDB reports for the five structures uploaded to the PDB, but table S1 should be checked for mistakes: I think that the unit cell c axis of 8WZT is not 81.7? And “a = b = c” should be changed to “a, b, c”, the same for the angles. And the R-work/R-free format of 8WZT should be corrected.

Reply 2-5C: We have already deposited five structures on the wwPDB server; however, their release is on hold for publication in case we receive any correction requests from referees. Therefore, they are not available in the web server yet. We are now attaching the PDB validation reports for the deposited PDB files.

We corrected the unit cell and R-work/R-free format along with the c-axis for 8WZT in the revised manuscript.

Minor formal issues:

- Line 91: “...during the CO release reaction could be characterized [word missing?] TR-SFX.”
- Line 135: MLCT should be spelled in full as metal-to-ligand charge-transfer
- Line 397: “it is believed that this [word missing?] possible because of the asymmetric protein environment.”
- Supporting Information, line 27: were purchased
- Supporting Information, line 113: “...and deposited [words missing?] wwPDB server...”

Reply: We thank the reviewer for pointing out these small mistakes which are corrected in the revised manuscript. We have corrected them one by one according to your suggestions.

Reviewer #3:

In the manuscript titled “Real-time observation of a metal complex-driven reaction intermediate using a porous protein crystal and serial femtosecond crystallography” by Maity et al. the authors present a study on a CO-releasing manganese complex by incorporation into lysozyme crystals and probing the time-resolved reaction by serial femtosecond crystallography at SACLA.

Presented is an interesting approach for facilitating the TR-study of a small molecule by incorporation into a large unit cell protein crystal, yielding indexable diffraction patterns with serial “snapshot” data collection at an FEL. The results from the electron density interpreted reaction intermediates are then further analyzed with QM/MM simulations. The technique itself is innovative and interesting and, with improved experimental rigor, would be worthy of publication in Nature Communications.

Some details regarding the experimental conditions are lacking, which could potentially impact the interpretation of results and conclusions drawn. Furthermore, the authors elude to inconsistencies in time points, again without sufficient experimental detail to evaluate their reasoning for this. Detailed concerns and suggestions are outlined below, along with minor comments.

Major concerns:

R3-1A: More detailed analysis on whether the interleaved dark structures for time points $>17\text{ms}$ is required based on the experimental conditions.

Reply3-1A: We agree with the reviewer that time points $>17\text{ms}$ could provide additional information on the later stage of the reaction. However, in the current experimental setup, it is hard to achieve $>17\text{ms}$ data points. As discussed in the manuscript, at longer time points (Figure S10), the observed difference maps were not in perfectly ascending order compared to time points 10ns, 100ns, and $1\mu\text{s}$. This supports the idea that the formation of various intermediates is possible since the energy barriers are a little higher in some cases, as shown in Figure S17. Therefore, we think determining structures at $>17\text{ms}$ complicates the assignment of intermediates. Instead, we performed a power titration by varying the laser intensity at two different delay times of 10 ns and $1\mu\text{s}$ (Page 14-15, Line 269-278 and Figure S9A and S9B). The results show a linear relationship

of difference density features at the initial stages of the reaction. This suggests we can identify only the highest populated structure at later stages.

R3-1B. What were the flowrates, jet diameter etc.? In a recent study by Li et al. (<https://doi.org/10.1107/S2052252521002177>), the authors found that they severely underestimated the required sample flowrate to adequately avoid light contamination in a similar high-viscosity jet set-up at SACLA.

Reply 3-1B: The flow rate was 2.4 $\mu\text{l}/\text{min}$, and the nozzle diameter was 75 μm . Such details are now included in the revised Supporting Information (Page S3, Line 68-70) and in the main text (Page 9, Line 160-162).

Regarding the comment associated with the paper by Li et al., which described the correlation between light contamination and sample flow rate, we would like to respond with the following:

(i) The interleaved dark data were measured between two light data sets (see the setting in Fig-4b). Therefore, there is a possibility of light contamination in the interleaved dark data, which can be minimized or avoided by adjusting the flow rate, etc. In our case, we always considered the difference maps obtained against the complete darkness data without laser to avoid light contamination. We added this in the revised SI (Page S4, Line 118-121). We used difference maps with interleaved dark data as a control to see the extent of light contamination. Only the data 1 $\mu\text{s}/1 \text{ ms}$ -20 μJ (Figure S6 and S10a) are affected by some light contamination. We further checked this by negative delay measurement (illumination by a pump laser after an XFEL pulse) with the shortest delay data, 10 ns (Figure S6c). This is also mentioned in SI (Page S4, line 122-123).

(ii) We agree with the reviewer that flow rate optimization is necessary to avoid light contamination, as done by Li et al. In our case, we used an already optimized setup as reported before (Science, 2016, 354(6319):1552-1557), and the setup has been followed in several works. Therefore, we performed no further experiment by varying the flow rate. The specifications are given in the XFEL measurement section in SI. Here are the specifications: Flow rate: 2.4 $\mu\text{l}/\text{min}$; jet diameter: 70 μm ; Laser spot size: 100 μm (FWHM).

R3-2: If assuming similar condition for the experiment presented here to that published by Li et al. (since these are missing from this manuscript); i.e. Jet diameter: 125 μm , while collecting at 30Hz: For the stated laser spot size of 100 μm FWHM, an absolute theoretical minimum flowrate

of 3 $\mu\text{L}/\text{min}$ would have been required to avoid light contamination. However, when considering the observations by Li et.al. that the optical illumination area is scattered well beyond the area defined by the laser spot size, they required a flowrate of $9.8\mu\text{l}/\text{min}$ to sufficiently replenish sample in between pulses with a 250 μm laser spot size (top-hat, not FWHM!). So I would expect similar requirements for the data presented here.

Reply3-2: Please see the last part to the reply R3-1B.

R3-3: How were the longer time delays achieved? By offsetting pump and probe in space? What were the parameters to achieve this? Was jet speed measured or estimated?

Reply3-3: The longest time delay, 17 ms, was achieved by raising the position of the pump laser 150 μm from the intersection point with XFELs. We set the pump laser 16.33 ms after the first XFEL shot of our 3-shots train (condition-1 (Cond. 1), condition-2 (Cond. 2), and condition-3 (Cond. 3) as initial setting). The Cond. 2 became the shot with a 17 ms delay after the pump laser (see the below Figure). The flow rate stayed the same at $2.4\mu\text{l}/\text{min}$, which equals an estimated $9.06\text{ mm}/\text{sec}$ jet speed with a 75 μm diameter injector. Based on the $110\mu\text{m}(\text{V}) * 90\mu\text{m}(\text{H})$ size of the pump laser and the aforementioned jet speed, those illuminated crystals arrived at the XFEL exposure point (Cond. 2) after $10.5\text{ ms} \sim 22.6\text{ ms}$. The target sample, which was 17 ms away from the XFEL exposure point (Cond. 2), was covered by the pump laser.

R3-4: The 1ms delay difference maps presented seems to be light contaminated (Figure S9b). The 17ms does not (Figure S9c).

Reply3-4: We agree that the interleaved dark-2 data for 1 ms might get some light contamination as it showed reduced difference density (CO-ax) in Figure S10 (previous Figure S9). Since the interleaved dark data were collected between two light data sets (Please see the setting in Figure 4b), light contamination is possible. Therefore, we always considered the difference map against complete darkness data in which no laser pulse was used. Difference map against interleaved dark data was used as a control to check the extent of light contamination. We explained this in the structure analysis section in SI (Page S4, Line 118-121). In addition, we also performed the experiment with a negative delay to check light contamination (for the shortest delay, 10ns) in which light was irradiated after an XFEL pulse (Figure S6b and S6c).

R3-5: Was the crystal suspension not washed prior to the SFX experiment, as for the UV-vis and ATR-IR spectral analysis? If not, what was the rationale if it was deemed necessary for the spectral analysis? Without sufficiently describing the experimental conditions this reviewer finds it hard to accept the authors' speculation that the inconsistencies in electron density result from replenishing compound from the bulk solvent. Light contamination seems more likely.

Reply3-5: The spectral measurements were performed to confirm the binding of the Mn-Carbonyl complex into the crystal. In this case, if the buffer contains a free metal complex, it might interfere with the spectrum of metal-bounded protein crystals. Thus, the crystals were washed before spectral measurement.

For X-ray diffraction measurement, the crystals were not washed, the soaking solution was removed, and the crystals were mixed directly with the high-viscosity grease. This is mentioned in the experimental section in SI (Page S3, Line 71-77). Since the crystals were coated with high-viscosity grease, we speculated that the released Mn might not escape out of the crystal but instead could rebound again. That could be why inconsistencies in the ascending order in the difference electron density at longer delay times. We explained this in the revised manuscript (Page 15, Line 286-288).

Light contamination is a different phenomenon and not associated with crystal preparation. It dependent on laser setting like intensity, delay time, flow rate etc. Please see the reply R3-1B.

R3-6: How were the 2 different laser fluences chosen? Is it conceivable that 40uJ is heating the sample, resulting in the overall blurring of the 2Fo-Fc maps (Figure 4c) and higher B-factors for the protein and solvent (Table S1B)? This reviewer would have liked to see a more detailed correlation analysis of the QM/MM results (Energy barriers presented in Figure 5b) and the pumping scheme/laser fluence employed. How many photons per reaction center?

Reply3-6: According to Figure 4b, we used pre-optimized conditions for TR-SFX measurement in which 30Hz repetition was used for the XFEL pulse and 10Hz repetition was used for the pump-probe laser pulse.

We appreciate the reviewer for raising any issue regarding heating during light exposure, which is a point of interest for us as well as spectroscopists. Recently, Wolff AM et al. reported a new method of TR-SFX using temperature-jump (T-jump) techniques (Wolff, A.M. et al., *Nat. Chem.* 15, 1549–1558 (2023)). They estimated how much the temperature rose by calculation. According to the equation in the paper (shown below), the temperature rise value is proportional to absorbance (absorption coefficient) and irradiation energy. A_{365} corresponds to approximately 10^{-5} of A_{1443} (https://water.lsbu.ac.uk/water/water_vibrational_spectrum.html). In the case of 40 μ J, the irradiation energy is ca. 0.074 of the energy used in the paper. Therefore, 21 K (estimated temperature rise in the paper) $\times 10^{-5} \times 0.074 = 1.56 \times 10^{-5}$ K. Thus, we concluded that heating is negligible in our experiments.

1. $A_{1443} = 12.83$ for water based on ref. 32; this corresponds to an attenuation depth of $\sim 780 \mu\text{m}$.

2. per Beer's law $\varepsilon = \frac{A_{1443}}{l \times M} = \frac{12.83}{1.000 \text{ cm} \times 55.56 \text{ M}} = 0.2309 \text{ cm}^{-1} \text{ M}^{-1}$

3. $A_{75\mu\text{m}} = 0.2309 \text{ cm}^{-1} \text{ M}^{-1} (7.5 \times 10^{-3} \text{ cm})(55.56 \text{ M}) = 0.096$

4. $q = I \times A_{75\mu\text{m}} = 5.4 \times 10^{-4} \text{ J} \times 0.096 = 5.2 \times 10^{-5} \text{ J}$

5. Volume = $r^2 \times \pi \times l = 5.0 \times 10^{-3} \text{ cm} \times 5.0 \times 10^{-3} \text{ cm} \times \pi \times 7.5 \times 10^{-3} \text{ cm} = 5.9 \times 10^{-7} \text{ cm}^3 = 5.9 \times 10^{-7} \text{ ml}$

6. $m = 5.9 \times 10^{-7} \text{ ml} \times 1.0 \text{ g ml}^{-1} = 5.9 \times 10^{-7} \text{ g}$

7. $\Delta T = \frac{q}{m \times c} = \frac{5.2 \times 10^{-5} \text{ J}}{5.9 \times 10^{-7} \text{ g} \times 4.184 \text{ J g}^{-1} \text{ K}^{-1}} = 21 \text{ K}$

Direct comparisons of the bond distances around the Mn center and root-mean-square deviation (RMSD) among the QM/MM optimized structures and the SFX results are shown in Table S5 and Table S6. The average Mn-C coordination length of the QM/MM optimized structure **1** is 1.81 Å, slightly shorter compared to the corresponding one of 1.94 Å in SFX result (PDBID:8WZF). The average Mn-O coordination length in QM/MM structure **1** is 2.12Å corresponding to the SFX (PDBID:8WZF) result of 2.24 Å. In both cases, the difference between QM/MM and SFX results is around -0.12 Å, and the trend is very similar. From the RMSD values around the Mn center (Table S6), the closest SFX structures from QM/MM structures (**1-3**) are commonly complete dark SFX model, whereas the closest QM/MM structures from SFX structures are **1**, **2** and **3** for SFX structures of (8WZF) complete dark, (8WZG) 10ns 20μJ (8WZV) 1μs 40μJ, respectively. Larger RMSD values of around 0.2 Å come from the larger disorder at the equatorial water ligands, not CO ligands. This implies the flexibility of the water coordination. The RMSD results indicate that structural similarity is highest in the initial non-photoexcited state, and the structural similarity becomes relatively worse as the reaction proceeds after the photoexcitation. It should be emphasized that the closest QM/MM structure in actual species corresponds best to the SFX structure. Based on these detailed structural comparisons, SFX structures reflect the photochemical reactions.

From the viewpoint of the energy, the relation between the reaction rate and the energy barrier is related as the Eyring equation

$$k = \frac{k_B T}{h} \exp\left(-\frac{\Delta E^\ddagger}{RT}\right)$$

where k_B is Boltzmann's constant, T is temperature, h is Planck's constant, ΔE is reaction energy, and R is the gas constant. As described above, we can assume $T=300\text{K}$, a reaction step with an energy barrier of $\Delta E^\ddagger=6.0 \text{ kcal mol}^{-1}$ occurs at a speed of $k=2.7 \times 10^8 \text{ sec}^{-1}$, which corresponds to a time rate of $1/k = 3.7 \text{ n sec}$. A reaction with an energy barrier of $\Delta E^\ddagger=9.2 \text{ kcal mol}^{-1}$ occurs at a speed of $k=1.2 \times 10^6 \text{ sec}^{-1}$, corresponding to a time rate of $1/k = 0.80 \mu \text{ sec}$. QM/MM calculated energy barriers for **1**->**2** and **2**->**3** are $6.0 \text{ kcal mol}^{-1}$ and $10.2 \text{ kcal mol}^{-1}$. Therefore, theoretically evaluated energy barriers correspond to the reactions of 10ns and 1μs delays, respectively.

Above discussion was added as a brief sentence in the main text (Page 14, line 269-278) and as a full discussion in the supporting information (Page S6-S7, line 165-176, and 194-212).

The blurring of the 2Fo-Fc map is possibly caused by the changes in the coordination structure of Mn when two COs were released. That causes disorder in the resulting structure, giving a higher B-factor and blurring the 2Fo-Fc map.

Based on the laser intensity and wavelength used, the number of photons per pulse was 3.85×10^{12} (for 20 μ J), which ~ 6.1 photon/metal site. Since the laser intensity should be reduced by high-viscous carrier media, the number of photons seems not excessive.

R3-7: Why is there such a noticeable discrepancy in the unit cell dimensions across the datasets (TableS1A, 80.8, 80.8, 37.6 for 1 us 40uJ; 81.0, 81.0, 37.7 for Complete Darkness and 10 ns 20uJ; 81.7, 81.7, 37.9 for 100ns 20uJ and 1us 20uJ)? Isomorphous difference maps are very sensitive to even minute differences in unit cell dimensions. Since the interleaved “dark” datasets match their respective “light” counterparts (and assuming they are not light contaminated) this is unlikely caused by the reaction; is it conceivable that the injector was not always installed in the same position (in which case the detector to sample difference should probably be tweaked instead of the differing unit cell constants)?

Reply3-7: The datasets were collected over three sessions. We added the data collection dates to the revised crystallographic Table S1A (Row 3) and Table S1B-F (Row 2). The differences probably arose from systematic errors in the detector geometry and/or batch-to-batch variations in crystal preparations. Because SACLA does not collect calibration-grade powder patterns before a beam time, we do not know the absolute detector distance. The geometry was optimized for each session to maximize the indexing rate and the symmetry of unit cell distributions. To avoid artifacts in difference maps, difference maps were calculated only between datasets collected on the same beam time.

R3-8: Some of the conclusions are over-generalized blanket statements, and the reader would benefit from more concrete specifications for this to be a widely applicable new method:

Reply 3-8: Thank you for pointing this out, we rechecked the paper and provided concrete specifications in the respective section (Page5, line 109-110; page24, line 449-456, page 25, line 483-484).

R3-9:- 22Å is not very large for a protein solvent channel. What implications does this have on systems that can be studied using HEWL.

Reply3-9: We agree with your concern. Other protein templates with larger solvent channels can be used for larger molecules. The explanation was added to the revised manuscript (Page 24, Line 449-456).

R3-10: HEWL was crystallized in its tetragonal isoform, despite the orthorhombic form having larger solvent channels. The authors rationalize this with the tetragonal form being easier to crystallize homogeneously with high yield. This is a reasonable argument when HEWL crystals are regarded as a readily accessible scaffold material. However, then arguing that the ligand environment can be easily modified by mutating the surrounding amino acids completely contradicts this. Cloning, recombinant expression, purification and then crystallizing a protein (even lysozyme) is a tedious and expensive endeavor. This reviewer would be more interested in the generalizability of this approach. E.g. what ligands would be amendable to studying using this technique. Or is it the uniqueness of this particular reaction (“After CO release, the Mn center will typically undergo dimerization, aerial oxidation or precipitation in solution and such results complicate mechanistic investigations”) that make it amendable to this technique? Are there a larger group of compounds that would bind to HEWL and lend itself to TR-SFX study?

Reply 3-10: We appreciate your understanding of our rationale for using HEWL crystals. In the previous manuscript, we mentioned amino acid substitutions as one possible generalization of our method using protein crystals. This is also related to the large supply of nanocrystals essential for the current requirement of the TR-SFX measurements. We have reported the synthesis of protein and mutant nanocrystals using intracellular crystallization and their immobilization of metal complexes (<https://doi.org/10.1039/C7CP06651A>). Using that method, the problems the referee pointed out can be overcome. Therefore, we are conducting them for TR-SFX research using the technique. However, since these are beyond the subject of this manuscript, they are only added as refs in the revised manuscript (Page 24, Line 449-456).

Minor comments:

R3-11: I'm not sure what the value of Figure 1a is. As ubiquitous as tetragonal HEWL crystals are, and protein crystals having solvent channels being common knowledge, this figure, at best, belongs in the supplement. Figure 3 has the same information, plus the more interesting depiction of the precise binding site of $\text{Mn}(\text{CO})_3$ to the HEWL molecules, and the associated accessibility via the solvent channel.

Reply3-11: We agree with the referee's observation that protein crystals with solvent channels are commonly known. Figure 1a represents the lattice packing in a metal-free state, whereas Figure 3c is our experimental result showing the metal-bounded state of HEWL. Considering a general understanding of the broad readership of our manuscript, we think keeping both images useful for readers from another background. As the reviewer pointed out and found similarities between the two panels, we revised the figure 1a a little and the caption (Page 6, line 113-115) and included additional texts highlighting the position of His15 and metal binding (Page 4, Line 86-87, and 89).

R3-12: Figure 2: Comparison spectra for apo-HEWL and post-photolysis would make for a nice addition. Also, c) has "au" as y-axis units whereas figure b) doesn't have units. The distinction may be arbitrary, but consistency would be nice.

Reply 3-12: Figure 2b and 2c have been corrected with addition of spectra for apo-HEWL and post photolysis.

R3-13: Figure 3c) replace the 20 "uJ" with " μJ "

Figure 5b) what are the units of delta E?

Reply3-13: The μJ has been corrected in Figure 4c. The unit of delta E is kcal/mol. This has been added in the caption of Figure 5 (page 22, line 410).

R3-14: Reference 33 is an odd unconventional choice. I suggest citing the original work by Chapman et al (<https://doi.org/10.1038/nature09750>) and Neutze et al. (<https://doi.org/10.1038/35021099>).

Reply 3-14: The new references (35 and 36) have been updated in the revised manuscript.

R3-15: Supplement: XFEL Data processing: What is meant by “merging strategies were optimized”? What parameters and sub-programs were ultimately used for the data presented? Were Friedel’s merged etc.?

Reply 3-15: The optimization strategy is described in Supporting Information (Page S3, line 85-90) as: “For each session, merging strategies were optimized by testing no scaling, scaling in process_hkl and scaling in partialator, while the per-image resolution limit by the push-res parameter was varied”. The resulting choices were added as: “The resulting strategies were scaling by partialator without partiality correction, push-res=0.75 for datasets collected in 2021 November and 2.00 (2024 February) or Monte Carlo integration by process_hkl without scaling, push-res=0.75 (2020 November) and 1.25 (2022 October).” Friedel pairs were merged.

R3-16: Figure S7: Why would the manganese occupancy differ from the COs, unlike for Figure S8 where they are consistent with the COs and the H₂O_s? Either the MnCO₃ complex is bound in all binding sights or not 10ns after illumination. This does not match the discussion that the COs are consecutively replaced by H₂O_s.

Reply 3-16: We appreciate you pointed it out. This mistake was corrected in the revised Supporting Information (Figure S7 (Column 2 in both a and b)).

R3-17: Language: While readable and understandable, the manuscript would benefit from a general language improvement with more concise terminology, e.g.:

P 3, line 59: “over a wide time range from femtoseconds” -> To? From femtoseconds is not a range.

P7, Line 120 and P 9, Line 152: “A single diffraction was collected per crystal” -> A single diffraction pattern was collected per crystal. Plus, this is not substantiated without stating flow rates!

Reply: This has included on page 7, line 124-125, page 9, line 159 and 160- 161.

Page 9, Line 153: “and a number of diffraction was collected” -> a minimum of XYZ indexable diffraction patterns were collected per dataset

Reply: This correction has been included on page 9, line 161-162.

Page 17, Line 208: “this suggests that the axial CO ligand is more sensitive to light” -> A CO ligand is not light-sensitive to my knowledge.

Reply: This has been corrected in the revised manuscript (Page 12, line 215-216). “This suggests that the axial.....” has been corrected to “This suggests that the Mn-COax bond exhibited greater susceptibility to light and thereby initiating the release of COax during this phase of the reaction”.

REVIEWERS' COMMENTS

Reviewer #2 (Remarks to the Author):

The revised manuscript has improved. I also appreciate that the experimental design is explained and verified in more detail. I still have minor doubts about an immediate broad applicability of the paper content at this stage of elaboration. The experimental concept (using a protein environment to structurally access the primary route of one specific reaction of an immobilized metal center) is however really valuable and provides the significance of the paper in comparison to previous time-resolved crystallography approaches. I consequently recommend the manuscript for publication.

Reviewer #3 (Remarks to the Author):

The authors have done a great job in revising the manuscript, in particular the addition of the laser fluence titration study is highly beneficial for supporting the manuscripts structural interpretations. Furthermore, the inclusion of a detailed comparison of the SFX structural results with the QM/MM simulation results make it a lot easier to comprehend and follow the authors' reasonings and conclusions.

Lastly, the additional detail regarding how the experiments were performed as well as the detailed responses and calculations in the rebuttal to initial experimental concerns is greatly appreciated and convincing.

I believe the manuscript is now suitable for publication in nature communications, pending 2 minor comments:

page 9 line 161: "XYZ" was intended as a place holder for the actual number patterns included in the respective datasets, and should be populated with such!

page 24 line 452: "Although the supply [...] could be addressed using in-cell protein crystals" I do not understand how this in any way addresses the problem! How would ligands be inserted into in-cell crystals? Furthermore, growing in vivo crystals is far more cumbersome and low-yield than in vitro crystallization!

Typo in Figure 4c: "Drak" should be "Dark"

Response to the reviewers

Reviewer #2 (Remarks to the Author):

Comment: The revised manuscript has improved. I also appreciate that the experimental design is explained and verified in more detail. I still have minor doubts about an immediate broad applicability of the paper content at this stage of elaboration. The experimental concept (using a protein environment to structurally access the primary route of one specific reaction of an immobilized metal center) is however really valuable and provides the significance of the paper in comparison to previous time-resolved crystallography approaches. I consequently recommend the manuscript for publication.

Response: We greatly appreciate your positive feedback on our revised manuscript. To address the immediate broad applicability of our work, protein crystals can be engineered to fix synthetic metal complex catalysts which can allow the study of catalytic reaction mechanisms by capturing real-time intermediate formation. Additionally, using serial crystallography, artificial metalloenzyme crystals can be used to explore their reaction mechanisms. Moreover, our approach can be extended to study other enzymatic processes and synthetic catalysts, providing insights into their functional dynamics. We believe these points illustrate the broader applicability and significance of our work.

Reviewer #3 (Remarks to the Author):

The authors have done a great job in revising the manuscript, in particular the addition of the laser fluence titration study is highly beneficial for supporting the manuscripts structural interpretations. Furthermore, the inclusion of a detailed comparison of the SFX structural results with the QM/MM simulation results make it a lot easier to comprehend and follow the authors' reasonings and conclusions.

Lastly, the additional detail regarding how the experiments were performed as well as the detailed responses and calculations in the rebuttal to initial experimental concerns is greatly appreciated and convincing.

I believe the manuscript is now suitable for publication in nature communications, pending 2 minor comments:

Response: We greatly appreciate your positive feedback on our revised manuscript.

Comment-1: page 9 line 161: "XYZ" was intended as a place holder for the actual number patterns included in the respective datasets, and should be populated with such!

Response: The sentence has been corrected and "XYZ" has been replaced by "15,000" (page 7, line 143).

Comment-2: page 24 line 452: "Although the supply [...] could be addressed using in-cell protein crystals" I do not understand how this in any way addresses the problem! How would ligands be inserted into in-cell crystals? Furthermore, growing in vivo crystals is far more cumbersome and low-yield than in vitro crystallization!

Response: Thank you for your comment on in-cell crystal. To avoid any confusion, we removed the sentence from main manuscript.

Comment-3: Typo in Figure 4c: "Drak" should be "Dark"

Response: The text "Drak" in Figure 4c has been corrected to "Dark".